# Asymptotics of Wide Networks from Feynman Diagrams

**Ethan Dyer & Guy Gur-Ari**[*]
Google
Mountain View, CA
{edyer,guyga}@google.com

## Abstract

Understanding the asymptotic behavior of wide networks is of considerable interest. In this work, we present a general method for analyzing this large width behavior. The method is an adaptation of Feynman diagrams, a standard tool for computing multivariate Gaussian integrals. We apply our method to study training dynamics, improving existing bounds and deriving new results on wide network evolution during stochastic gradient descent. Going beyond the strict large width limit, we present closed-form expressions for higher-order terms governing wide network training, and test these predictions empirically.

## 1 Introduction

Neural networks achieve remarkable performance on a wide array of machine learning tasks, yet a complete analytic understanding of deep networks remains elusive. One promising approach is to consider the large width limit, in which the number of neurons in one or several layers is taken to be large. In this limit one can use a mean-field approach to better understand the network's properties at initialization (Neal (1996); Lee et al. (2018)), as well as its training dynamics (Daniely (2017); Jacot et al. (2018)). Additional related works are cited below.

Suppose that $f(x)$ is the network function evaluated at an input $x$. Let us denote the vector of model parameters by $\theta$, whose elements are initially chosen to be i.i.d. Gaussian. In this work we consider a class of functions we call *correlation functions*, obtained by taking the ensemble averages of $f$, its products, and its derivatives with respect to the parameters $\theta$, evaluated on arbitrary inputs. Here are a few examples of correlation functions.

$$\mathbb{E}_\theta \left[ f(x_1) f(x_2) \right] , \quad \sum_\mu \mathbb{E}_\theta \left[ \frac{\partial f(x_1)}{\partial \theta^\mu} \frac{\partial f(x_2)}{\partial \theta^\mu} \right] , \quad \sum_{\mu,\nu} \mathbb{E}_\theta \left[ \frac{\partial f(x_1)}{\partial \theta^\mu} \frac{\partial f(x_2)}{\partial \theta^\nu} \frac{\partial^2 f(x_3)}{\partial \theta^\mu \partial \theta^\nu} f(x_4) \right] . \quad (1)$$

Correlation functions often show up in the study of wide networks. For example, the first correlation function in (1) plays a central role in the Gaussian Process picture of wide networks (Lee et al. (2018)), and has been used to diagnose signal propagation in wide networks (Pennington et al. (2017)). The second example in (1) is the ensemble average of the Neural Tangent Kernel (NTK), which controls the evolution of wide networks under gradient flow (Jacot et al. (2018)), and the third example shows up when computing the time derivative of the NTK with MSE loss.

While correlation functions can be computed analytically in some special cases (Cho & Saul (2009)), they are not analytically tractable in general. In this work, we present a method for bounding the asymptotic behavior of such functions at large width. In many cases of interest, the bound we obtain is tight. Derivation of the method relies on Feynman diagrams (Feynman (1949)), a technique for calculating multivariate Gaussian integrals, and specifically on the 't Hooft expansion ('t Hooft (1974)). However, applying the method is straightforward and does not require any knowledge of Feynman diagrams.

**Motivation.** The method presented here is a general-purpose tool that can be added to a researcher's toolbox. It allows a researcher to quickly identify ways in which network behavior and training

---

[*]Authors listed alphabetically.

dynamics simplify at large width, for example by deriving which quantities vanish in this limit. As we demonstrate, using this method one is able to dramatically cut down on the amount of effort required to derive several key results related to wide networks, as well as to go beyond the infinite width limit and investigate ordinary neural networks.

**Our contribution.**

1. We present a general method for bounding the asymptotic behavior of *correlation functions*. The method is an adaptation of Feynman diagrams to the case of wide neural networks. The adaptation involves a novel treatment of derivatives of the network function, an element that is not present in the original theoretical physics formulation.

2. We apply the method to the study of wide network evolution under gradient descent. We improve on existing results for gradient flow (Jacot et al. (2018)) by deriving tighter bounds, and extending the analysis to the case of stochastic gradient descent (SGD). Going beyond the infinite-width limit, we present a formalisn for deriving finite-width corrections to network evolution, and present explicit formulas for the first order correction. To our knowledge, this is the first time this correction has been calculated.

3. As additional applications of our method, in Appendix E.2 we show that in the large width limit the SGD updates are linear in the learning rate, and in Appendix E.3 we discuss finite width corrections to the spectrum of the Hessian.

**Limitations of our approach.**    The main result of this paper is a conjecture. We test our predictions extensively using numerical experiments, and prove the conjecture in some cases, but we do not have a proof that applies to all the cases we tested, including for deep networks with general non-linearities. Furthermore, our method can only be used to derive asymptotic bounds at large width; it does not produce the width-independent coefficient, which is often of interest.

**Related work.**    For additional works on wide networks, including relating them to Gaussian processes, see de G. Matthews et al. (2018); Novak et al. (2019); Garriga-Alonso et al. (2019); Yang (2019); Pennington & Bahri (2017); Pennington & Worah (2018); Schoenholz et al. (2017); Xiao et al. (2018); Chen et al. (2018); Daniely et al. (2016); Lee et al. (2019); Du et al. (2018b;a); Allen-Zhu et al. (2018). For additional works discussing the training dynamics of wide networks see Geiger et al. (2019a); Arora et al. (2019). For a previous use of diagrams in this context, see Pennington & Worah (2017). The Neural Tangent Hierarchy presented in Huang & Yau (2019), published during the completion of this version, has significant overlap with the recursive differential equations (10) presented below.

The rest of the paper is organized as follows. In Section 2 we present our main conjecture and supporting evidence. In Section 3 we apply the method to gradient descent evolution of wide networks, and in Section 4 we present details on Feynman diagrams, which is the basic technique used in our proofs. We conclude with a Discussion. Proofs, additional applications, and details can be found in the Appendices.

**Note**: An earlier version of this work appeared in the ICML 2019 workshop, Theoretical Physics for Deep Learning (Dyer & Gur-Ari (2019)).

## 2    CORRELATION FUNCTION ASYMPTOTICS

In this section we present our main result: a method for computing asymptotic bounds on correlation functions of wide networks. We present the result as a conjecture, supported by analytic and empirical evidence.

### 2.1    NOTATION

Let $f(x) \in \mathbb{R}$ be the network output of a deep network with $d$ hidden layers and input $x \in \mathbb{R}^{D_{\text{in}}}$, defined by

$$f(x) = n^{-1/2} V^T \sigma(n^{-1/2} W^{d-1} \cdots \sigma(n^{-1/2} W^1 \sigma(Ux))). \tag{2}$$

Here $U \in \mathbb{R}^{n \times D_{\text{in}}}$, $V \in \mathbb{R}^n$, $W^1, \ldots, W^{d-1}$ are weight matrices of dimension $n$, and $\sigma : \mathbb{R} \to \mathbb{R}$ is the non-linearity.[1] We denote the vector of all model parameters by $\theta$. At initialization, the elements of $\theta$ are independent Gaussian variables, with each element $\theta^\mu \sim \mathcal{N}(0, 1)$. The corresponding distribution of $\theta$ is denoted by $\mathcal{P}_0$.[2]

Let us now define *correlation functions*, the class of functions that is the focus of this work. These functions involve derivative tensors of the network function. We denote the rank-$k$ derivative tensor by $T_{\mu_1 \ldots \mu_k}(x; f) := \partial^k f(x)/\partial\theta^{\mu_1} \cdots \partial\theta^{\mu_k}$. For $k = 0$ we define $T(x; f) := f(x)$, and still refer to this as a derivative tensor for consistency.

**Definition 1.** *A* correlation function *is the expectation value of a product of derivative tensors, evaluated at arbitrary inputs, where the tensor indices are summed in pairs over all the model parameters. A general correlation function $C$ takes the form*

$$C(x_1, \ldots, x_m) := \sum_{\mu_1, \ldots, \mu_{k_m}} \Delta^{(\pi)}_{\mu_1 \ldots \mu_{k_m}} \mathbb{E}_\theta \left[ T_{\mu_1 \ldots \mu_{k_1}}(x_1) T_{\mu_{k_1+1} \ldots \mu_{k_2}}(x_2) \cdots T_{\mu_{k_{m-1}+1} \ldots \mu_{k_m}}(x_m) \right].$$

(3)

*Here, $0 \le k_1 \le \cdots \le k_{m-1} \le k_m$ are integers,[3] $m$ and $k_m$ are even, $\pi \in S_{k_m}$ is a permutation, and $\Delta^{(\pi)}_{\mu_1 \ldots \mu_{k_m}} = \delta_{\mu_{\pi(1)} \mu_{\pi(2)}} \cdots \delta_{\mu_{\pi(k_m-1)} \mu_{\pi(k_m)}}$. We use $\delta$ to denote the Kronecker delta.*

If two derivative tensors in a correlation function have matching indices that are summed over, we say that they are *contracted*. For example, the correlation function $\sum_\mu \mathbb{E}_\theta \left[ \partial f(x_1)/\partial\theta^\mu \cdot \partial f(x_2)/\partial\theta^\mu \right]$ has one pair of contracted tensors. See (1) for additional examples of correlation functions.

## 2.2 Asymptotic bounds on wide networks

We now present our main conjecture, which allows us to place asymptotic bounds on general correlation functions of wide networks.

**Conjecture 1.** *Let $C(x_1, \ldots, x_m)$ be a correlation function. The* cluster graph $G_C(V, E)$ *of $C$ is a graph with vertices $V = \{v_1, \ldots, v_m\}$, where $v_i = T(x_i)$ for all $i$, and edges $E = \{(v_i, v_j) \mid (T(x_i), T(x_j))$ contracted in $C\}$. Suppose that the cluster graph $G_C$ has $n_e$ connected components with an even size (even number of vertices), and $n_o$ components of odd size. Then $C(x_1, \ldots, x_m) = \mathcal{O}(n^{s_C})$, where*

$$s_C = n_e + \frac{n_o}{2} - \frac{m}{2}.$$

(4)

We will refer to the connected components of a cluster graph $G_C$ as the clusters of $C$. Table 1 lists examples of bounds derived using the Conjecture for several correlation functions. The intuition behind Conjecture 1 comes from the following result for deep linear networks.

**Theorem 1.** *Conjecture 1 holds for correlation functions of networks with linear activations, including for deep linear networks with biases (affine transformations).*

Let us discuss the intuition behind this theorem. Computing correlation functions of deep linear networks amounts to evaluating Gaussian integrals with polynomial integrands in $\theta$. One can evaluate such integrals using Isserlis' theorem, which tells us how to express moments of multivariate Gaussian variables in terms of their second moments. For example, given centered Gaussian variables $z_1, \ldots, z_4$,

$$\mathbb{E}_z [z_1 z_2 z_3 z_4] = \mathbb{E}_z [z_1 z_2] \mathbb{E}_z [z_3 z_4] + \mathbb{E}_z [z_1 z_3] \mathbb{E}_z [z_2 z_4] + \mathbb{E}_z [z_1 z_4] \mathbb{E}_z [z_2 z_3].$$

(5)

Therefore, correlation functions of deep linear networks can be expressed in terms of the covariances $\mathbb{E}_\theta [U_{i\alpha} U_{j\beta}] = \delta_{ij} \delta_{\alpha\beta}$, $\mathbb{E}_\theta [V_i V_j] = \delta_{ij}$, and $\mathbb{E}_\theta \left[ W^{(l)}_{ij} W^{(l)}_{kl} \right] = \delta_{ik} \delta_{jl}$. For example, for a deep linear network with 2 hidden layers, we have

$$\mathbb{E}_\theta [f(x_1) f(x_2)] = \frac{1}{n^2} \mathbb{E}_\theta \left[ V^T W U x_1 V^T W U x_2 \right] = \frac{x_1^T x_2}{n^2} \sum_{i,k}^n \delta_{ik} \delta_{ik} \sum_{j,l}^n \delta_{jl} \delta_{jl} = x_1^T x_2.$$

(6)

---

[1] We take all layers widths to be equal to $n$ for simplicity, but our results hold in the more general case where all widths scale linearly with $n$.

[2] We use $\mu, \nu, \ldots$ to denote $\theta$ indices, $i, j, \ldots$ to denote individual weight matrix and weight vector indices, and $\alpha, \beta, \ldots$ for input dimension indices.

[3] When $k_a = k_{a-1}$, the tensor $T(x_a)$ has no derivatives.

Every correlation function of a deep linear network can be similarly reduced to sums over products of Kronecker delta functions and width-independent functions of the inputs. The asymptotic large width behavior is determined by these sums over delta functions, which are tedious to compute by hand. Feynman diagrams are a graphical tool for computing these sums, allowing us to obtain the asymptotic behavior with minimal effort. This tool, which is described in detail in Section 4, is used to prove Theorem 1.

For networks with non-linear activations we further show the following

**Theorem 2.** *Conjecture 1 holds for (1) networks with ReLU activations, where all inputs are set to be equal, and for (2) networks with one hidden layer and smooth activation.*

For case (1), the idea behind the proof is to put an asymptotic bound on the ReLU network in terms of a corresponding deep linear network. For case (2), the basic idea is that each network function contains a single sum over the width, and by keeping track of these sums using Feynman diagrams we are able to bound the asymptotic behavior. We refer the reader to Appendix C for details.

## 2.3 NUMERICAL EXPERIMENTS

Table 1 lists asymptotic bounds on several correlation functions, derived using Conjecture 1. These are compared against the asymptotic behavior computed using numerical experiments. In addition to the results presented here, we performed experiments using the same correlation functions and experimental setup, but with weights sampled uniformly from $\{\pm 1\}$ instead of from a Gaussian distribution. The results are shown in Appendix A.1.

In all cases that were tested empirically, we found that Conjecture 1 holds. For networks with smooth activations, we found that Conjecture 1 always gives a tight bound. For networks with linear or ReLU activations, we always find that the Conjecture holds as an upper bound, but that sometimes the bound is not tight. One such case is highlighted in Table 1. In such cases, a tight bound can be obtained for deep linear networks using the complete Feynman diagram analysis presented below.[4]

| Correlation function $C$ | $n_e, n_o$ | $s_C$ | lin. | ReLU | tanh |
|---|---|---|---|---|---|
| $\mathbb{E}_\theta \left[ f(x_1) f(x_2) \right]$ | 0,2 | 0 | -0.02 | 0.003 | -0.02 |
| $\mathbb{E}_\theta \left[ f(x_1) f(x_2) f(x_3) f(x_4) \right]$ | 0,4 | 0 | -0.01 | 0.03 | -0.03 |
| $\sum_\mu \mathbb{E}_\theta \left[ \partial_\mu f(x_1) \partial_\mu f(x_2) \right]$ | 1,0 | 0 | 0.00 | 0.00 | 0.00 |
| $\sum_{\mu,\nu} \mathbb{E}_\theta \left[ \partial_\mu f(x_1) \partial_\nu f(x_2) \partial_{\mu,\nu} f(x_3) f(x_4) \right]$ | 0,2 | -1 | -0.98 | -1.03 | -1.01 |
| $\sum_{\mu,\nu,\rho} \mathbb{E}_\theta \left[ \partial_\mu f(x_1) \partial_\nu f(x_2) \partial_\rho f(x_3) \partial_{\mu,\nu,\rho} f(x_4) \right]$ | 1,0 | -1 | -2.01* | -2.01* | -0.98 |
| $\sum_{\mu,\nu,\rho,\sigma} \mathbb{E}_\theta \left[ \partial_\mu f(x_1) \partial_\nu f(x_2) \partial_{\mu,\nu} f(x_3) \partial_\rho f(x_4) \partial_\sigma f(x_5) \partial_{\rho,\sigma} f(x_6) \right]$ | 0,2 | -2 | -2.05 | -2.01 | -1.99 |

Table 1: Examples of bounds on correlation functions obtained from Conjecture 1. The 3 rightmost columns list numerical results for fully-connected networks with 3 hidden layers and with linear, ReLU, and tanh activations. The values listed in these columns are the fitted exponents for networks with the corresponding activations. The entries marked with an asterisk are those for which the predicted bound is not tight. The numerical results are obtained by computing the correlation functions for networks with widths $2^7, 2^8, \ldots, 2^{13}$, each averaged over 1,000 initializations, and fitting the exponent. Inputs are chosen to be random vectors of dimension 4.

## 3 APPLICATIONS TO TRAINING DYNAMICS

In this section we apply Conjecture 1 to study the evolution of wide networks under gradient flow and gradient descent. We begin by briefly reviewing existing results. Let $D_{\text{tr}}$ be a training set of size $M$,

---

[4]For deep linear networks, the proof of Theorem 1 relies on mapping diagrams to triangulations of 2D surfaces with at least 1 boundary. There are cases, such as the one highlighted in Table 1, in which all diagrams one can write down have at least 2 boundaries. In such cases, the full diagrammatic calculation gives a tight bound, while Conjecture 1 provides a valid upper bound that is not tight.

and let $L = \sum_{(x,y) \in D_{\text{tr}}} \ell(x, y)$ be the MSE loss, with single sample loss $\ell(x, y) = \frac{1}{2}(f(x) - y)^2$. The gradient flow equation is $\frac{d\theta}{dt} = -\nabla_\theta L$. The evolution of the network function under gradient flow is given by

$$\frac{df(x)}{dt} = -\sum_{(x',y') \in D_{\text{tr}}} \Theta(x, x') \frac{\partial \ell(x', y')}{\partial f} \, . \tag{7}$$

Here, $\Theta$ is the Neural Tangent Kernel (NTK), defined by $\Theta(x_1, x_2) := \nabla_\theta f^T(x_1) \nabla_\theta f(x_2)$. The authors of Jacot et al. (2018) showed that the kernel is constant during training up to $\mathcal{O}(n^{-1/2})$ corrections. This leads to a dramatic simplification in training dynamics (Jacot et al. (2018); Lee et al. (2019)). In particular, for MSE loss the network map evaluated on the training data evolves as $f(t) = y + e^{-t\Theta^{(0)}}(f^{(0)} - y)$.[5] We will use our technology to derive a tighter bound on finite-width corrections to the kernel during training and present explicit formulas for the leading correction.

The following result is useful in analyzing the behavior of correlation functions under gradient flow.

**Lemma 1.** *Let $C(\vec{x}) = \mathbb{E}_\theta [F(\vec{x})]$ be a correlation function, where $\vec{x} = (x_1, \ldots, x_m)$, and suppose that $C = \mathcal{O}(n^{s_C})$ for $s_C$ as defined in Conjecture 1. Then $\mathbb{E}_\theta \left[ \frac{d^k F(\vec{x})}{dt^k} \right] = \mathcal{O}(n^{s_C})$ for all k.*

Here we prove the statement for $k = 1$. Appendix D.2 contains a proof for the general case.

*proof (k = 1).* Let $C$ have $n_e$ even clusters and $n_o$ odd clusters. Consider $\mathbb{E}_\theta \left[ \frac{dF(\vec{x})}{dt} \right] = -\sum_\mu \sum_{x' \in D_{\text{tr}}} \mathbb{E}_\theta \left[ \frac{\partial F(\vec{x})}{\partial \theta^\mu} \frac{\partial f(x')}{\partial \theta^\mu} f(x') \right]$. Denote by $n'_e$ $(n'_o)$ the number of even (odd) clusters in this correlation function, which has $m' = m + 2$ derivative tensors. One can check that either $(n'_e, n'_o) = (n_e + 1, n_o)$ or $(n'_e, n'_o) = (n_e - 1, n_o + 2)$, depending on whether the $\partial_\mu F$ derivative is acting on an odd or even cluster in $F$.[6] Therefore, $n'_e + \frac{n'_o}{2} - \frac{m'}{2} \leq s_C$. $\qquad \square$

With this result, it is easy to understand the constancy of the NTK at large width. The first derivative of the NTK is given by

$$\mathbb{E}_\theta \left[ \frac{d\Theta(x_1, x_2)}{dt} \right] = -\sum_{x' \in D_{\text{tr}}}^M \sum_{\mu, \nu} \mathbb{E}_\theta \left[ \frac{\partial^2 f(x_1)}{\partial \theta^\mu \partial \theta^\nu} \frac{\partial f(x_2)}{\partial \theta^\mu} \frac{\partial f(x')}{\partial \theta^\nu} f(x') \right] + (x_1 \leftrightarrow x_2) . \tag{8}$$

Here, $(x_1 \leftrightarrow x_2)$ means "add the same term, but exchange $x_1$ and $x_2$". This correlation function has $n_e = 0$, $n_o = 2$, and $m = 4$, and is therefore $\mathcal{O}(n^{-1})$ by Conjecture 1. By Lemma 1, all higher-order time derivatives of the NTK are $\mathcal{O}(n^{-1})$ as well. If we now assume that the time-evolved kernel $\Theta(t)$ is analytic in training time $t$, and that we are free to exchange the Taylor expansion in time with the large width limit, then we find that $\mathbb{E}_\theta [\Theta(t) - \Theta(0)] = \sum_{k=1}^\infty \frac{t^k}{k!} \mathbb{E}_\theta \left[ \frac{d^k \Theta(0)}{dt^k} \right] = \mathcal{O}(n^{-1})$ for any fixed $t$. This bound, which is tighter than that found in Jacot et al. (2018), was noticed empirically in Lee et al. (2019) as well as in our own experiments, see Figure 1a.

This analysis can be extended to show that $\mathbb{E}_\theta [\Theta(t) - \Theta(0)] = \mathcal{O}(n^{-1})$ for SGD as well. The technique is similar, and again relies on Conjecture 1. We refer the reader to Appendix E.1 for details. These results improve on existing ones in several ways. Our method applies to the case of SGD as well as to networks where all layer widths are increased simultaneously — a setup that has proven to be difficult to analyze. In addition, the $\mathcal{O}(n^{-1})$ bound we derive on kernel corrections during SGD is empirically tight, and improves on the existing bound of $\mathcal{O}(n^{-1/2})$ which was derived for gradient flow (Jacot et al. (2018)).

---

[5] Here we are using condensed notation: $\Theta^{(0)}$ and $f^{(0)}$ are values at initialization, and $f, f^{(0)}, y$ are treated as vectors in training set space. The kernel $\Theta^{(0)}$ is a square matrix in the same space.

[6] Here we are extending the use of the term cluster to refer to derivative tensors in the integrand itself.

### 3.1 FINITE WIDTH CORRECTIONS

Next, we will compute the explicit time dependence of $\Theta$ and $f$ at order $\mathcal{O}(n^{-1})$ under gradient flow. This is the leading correction to the infinite width result. We define the functions $O_1(x) := f(x)$ and

$$O_s(x_1, \ldots, x_s) := \sum_\mu \frac{\partial O_{s-1}(x_1, \ldots, x_{s-1})}{\partial \theta_\mu} \frac{\partial f(x_s)}{\partial \theta_\mu}, \quad s \geq 2. \tag{9}$$

Notice that $O_2 = \Theta$ is the kernel. It is easy to check that

$$\frac{dO_s(x_1, \ldots, x_s)}{dt} = - \sum_{(x', y') \in D_{\text{tr}}} O_{s+1}(x_1, \ldots, x_s, x')(f(x') - y'), \quad s \geq 1. \tag{10}$$

We can use equations (9) and (10) to solve for the evolution of the kernel and network map. Notice that each function $O_s$ has $s$ derivative tensors and a single cluster. As a result, correlation functions involving operators with larger $s$ are increasingly suppressed in width. In particular, $\mathbb{E}_\theta\left[dO_4/dt\right] = -\sum_{x \in D_{\text{tr}}} \mathbb{E}_\theta\left[O_5(x)f(x)\right] = \mathcal{O}(n^{-2})$ at all times, using Conjecture 1 and Lemma 1. Thus to solve for $f$ and $\Theta$ at $\mathcal{O}(n^{-1})$ we can set $O_s = 0$ for all $s \geq 5$.

Let us denote the time-evolved kernel by $\Theta(t) = \Theta^{(0)} + \Theta_1(t) + \mathcal{O}(n^{-2})$, where $\Theta^{(0)}$ is the kernel at initialization, and $\Theta_1(t)$ is the $\mathcal{O}(n^{-1})$ correction we are seeking. Integrating equations (10) starting with $s = 4$, we find

$$\Theta_1(x_1, x_2; t) = - \int_0^t dt' \sum_{x \in D_{\text{tr}}} O_3^{(0)}(x_1, x_2, x) \Delta f(x; t')$$

$$+ \int_0^t dt' \int_0^{t'} dt'' \sum_{x, x' \in D_{\text{tr}}} O_4^{(0)}(x_1, x_2, x, x') \Delta f(x'; t'') \Delta f(x; t'). \tag{11}$$

Here we have introduced the notation $\Delta f(x; t) = e^{-t\Theta_0}(f^{(0)} - y)$. A detailed derivation can be found in Appendix E.4. There we also evaluate the integrals in (11) in terms of the NTK spectrum.

To obtain the $\mathcal{O}(n^{-1})$ correction to the network map (evaluated for simplicity on the training data), we further integrate (10) for $s = 1$ and find

$$f(t) = f_0(t) - e^{-t\Theta^{(0)}} \int_0^t dt' e^{t'\Theta^{(0)}} \Theta_1(t') e^{-t'\Theta^{(0)}}(f^{(0)} - y) + \mathcal{O}(n^{-2}). \tag{12}$$

Here we have denote the infinite width evolution by $f_0(t) = y + e^{-t\Theta^{(0)}}(f^{(0)} - y)$. Figures 1b and 1c compare these predictions against empirical results.

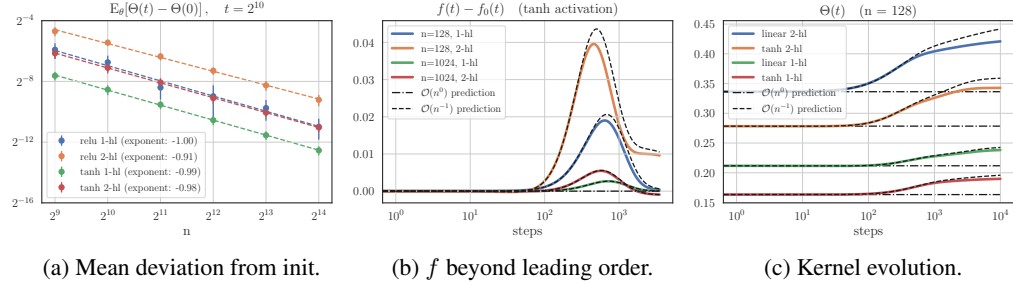

| (a) Mean deviation from init. | (b) $f$ beyond leading order. | (c) Kernel evolution. |

Figure 1: Empirical verification of predicted asymptotics. (a) The mean deviation of the NTK from its initial value for a variety of widths and activation functions. The fit (dashed) matches well with the predicted $\mathcal{O}(n^{-1})$ asymptotics. (b-c) Comparison between the empirical evolution (solid) and the $\mathcal{O}(n^{-1})$ predicted evolution (dashed) for the network function and the kernel. All experiments were performed on two-class MNIST, computing a single randomly-chosen component of $\Theta$ or $f$. See Appendix A for additional experimental details.

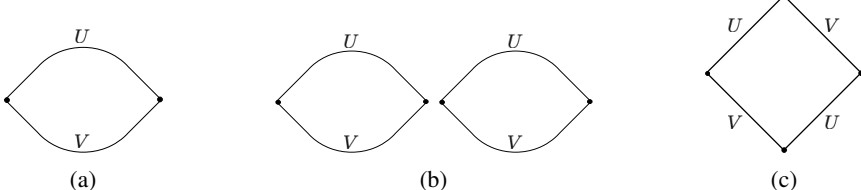

Figure 2: Feynman diagram examples. (a) The Feynman diagram of $\mathbb{E}_\theta\left[f(x)f(x')\right]$. (b)-(c) The Feynman diagrams of $\mathbb{E}_\theta\left[f(x_1)f(x_2)f(x_3)f(x_4)\right]$; additional, equivalent diagrams are not shown.

## 4 FEYNMAN DIAGRAMS FOR DEEP LINEAR NETWORKS

In this section we present the Feynman diagram technique, and show how it allows us to compute the asymptotic behavior of correlation functions. We end this section with a proof of Theorem 1 for the case of networks with a single hidden layer and linear activations.

Given a correlation function $C$, we map it to a family of graphs called Feynman diagrams. The graphs are independent of the inputs, and are defined as follows.

**Definition 2.** *Let $C(x_1,\ldots,x_m)$ be a correlation function for a network with $d$ hidden layers. The family $\Gamma(C)$ is the set of all graphs that have the following properties.*

1. *There are $m$ vertices $v_1,\ldots,v_m$, each of degree $d+1$.*

2. *Each edge has a type $t \in \{U, W^1,\ldots, W^{d-1}, V\}$. Every vertex has one edge of each type.*

3. *If two derivative tensors $T_{\mu_1,\ldots,\mu_\ell}(x_i), T_{\nu_1,\ldots,\nu_{\ell'}}(x_j)$ are contracted $k$ times in $C$, the graph must have at least $k$ edges (of any type) connecting the vertices $v_i, v_j$.*

*The graphs in $\Gamma(C)$ are called the* Feynman diagrams *of $C$.*

**Single hidden layer.** For the rest of this section we focus on networks with a single hidden layer. We refer the reader to Appendix B for a full treatment of deep linear networks. For networks with one hidden layer and linear activation, the network output is $f(x) = n^{-1/2}V^T U x$. Consider the correlation function $C(x_1, x_2) = \mathbb{E}_\theta\left[f(x_1)f(x_2)\right]$. We have

$$C(x_1, x_2) = \frac{1}{n}\sum_{i,j}^{n}\sum_{\alpha,\beta}^{D_{\text{in}}}\mathbb{E}_V\left[V_i V_j\right]\mathbb{E}_U\left[U_{i\alpha}U_{j\beta}\right]x_1^\alpha x_2^\beta = \frac{x_1^T x_2}{n}\sum_{i,j}^{n}\delta_{ij}\delta_{ij} = x_1^T x_2\,. \tag{13}$$

As the factors of $x_1$ and $x_2$ are independent of $n$, we see that $C(x_1, x_2) = \mathcal{O}(n^0)$. Notice that there are two relevant contributions to this answer: each factor of the network function in the integrand contributes $n^{-1/2}$, and the summed-over product of Kronecker deltas contributes $n$. Other details, such as the input dependence, are irrelevant. Feynman diagrams allow us to encode only those details that affect the $n$ scaling, ignoring the rest.

The set $\Gamma(C)$ for the correlation function (13) consists of a single Feynman diagram, shown in Figure 2a. The asymptotic bound on a correlation function is obtained by the following result, which is due to 't Hooft (1974).

**Theorem 3.** *Let $C(x_1,\ldots,x_m)$ be a correlation function with one hidden layer and linear activation. Then $C = \mathcal{O}(n^s)$ where $s = \max_{\gamma\in\Gamma(C)} l_\gamma - \frac{m}{2}$, and $l_\gamma$ is the number of loops in $\gamma$.*[7]

Let us give some intuition for Theorem 3. Each Feynman diagram $\gamma$ encodes a subset of the terms contributing to the correlation function. To get the asymptotic bound on the correlation function, we sum over the contributions of individual diagrams. We can compute the asymptotic behavior of a single diagram $\gamma$ using the following *Feynman rules*: (1) each vertex contributes a factor of $n^{-1/2}$, and (2) each loop contributes a factor of $n$. Therefore, if a diagram has $l_\gamma$ loops, its contribution to

---

[7]For networks with a single hidden layer, the number of loops in a diagram is equal to the number of connected components. This is not true for a general deep linear network.

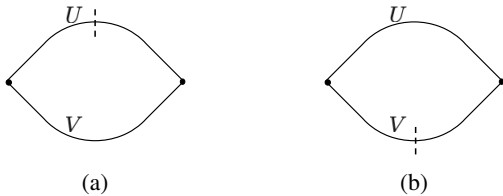

Figure 3: Feynman diagrams for $\mathbb{E}_{\theta}\left[\Theta(x, x')\right]$ with one hidden layer. The dashed vertical line represents vertices forced by contracted derivatives.

the correlation function scales as $n^{l_{\gamma}-m/2}$. Rule (1) is due to the explicit $n^{-1/2}$ factor in the network definition. Rule (2) follows from applying Isserlis' theorem, as follows. Each covariance factor (such as the factor $\mathbb{E}\left[V_i V_j\right] = \delta_{ij}$ in eq. (13)) corresponds to an edge in a Feynman diagram. A loop in the diagram corresponds to a sum over a product of Kronecker deltas, yielding a factor of $n$.

Returning to our example, the graph in Figure 2a has two vertices and one closed loop, so we recover the asymptotic behavior $\mathcal{O}(n^0)$ from Theorem 3. As another example, consider the correlation function $C(x_1, x_2, x_3, x_4) = \mathbb{E}_{\theta}\left[f(x_1)f(x_2)f(x_3)f(x_4)\right]$. It has two Feynman diagrams, shown in Figures 2b and 2c. Using the Feynman rules, we find that the disconnected graph represents terms that scale as $\mathcal{O}(n^0)$, while the connected graph represents terms that scale as $\mathcal{O}(n^{-1})$. Therefore, $C(x_1, x_2, x_3, x_4) = \mathcal{O}(n^0)$. This is an example of a more general phenomenon, that connected graphs vanish faster at large $n$ compared with disconnected graphs.

**Correlation functions with derivatives.** We now extend the Feynman diagram technique to correlation functions that include derivatives of $f$.[8] As a concrete example, consider the correlation function $C(x, x') = \mathbb{E}_{\theta}\left[\Theta(x, x')\right]$, where $\Theta$ is the kernel defined in Section 3. For a single hidden layer, we have

$$C(x, x') = \sum_{i=1}^{n} \mathbb{E}_{\theta}\left[\frac{\partial f(x)}{\partial U_i}\frac{\partial f(x')}{\partial U_i} + \frac{\partial f(x)}{\partial V_i}\frac{\partial f(x')}{\partial V_i}\right]. \tag{14}$$

The two derivative tensors in this correlation function are contracted: their indices are set to be equal and summed over. Therefore, according to Definition 2, $\Gamma(C)$ includes all diagrams in which the corresponding vertices share at least one edge. The resulting diagrams are shown in Figure 3. The edges forced by the contraction are explicitly marked by dashed lines for clarity, but mathematically they are ordinary edges. We see that in fact there is only one diagram contributing to this correlation function — the same one shown in Figure 2a. Following the Feynman rules, we find that $\mathbb{E}_{\theta}\left[\Theta(x, x')\right] = \mathcal{O}(n^0)$.

The fact that contracted derivatives should be mapped to forced edges in the Feynman diagrams is proved in Appendix B. The basic reason behind this rule is the relation $\sum_k \frac{\partial V_i}{\partial V_k}\frac{\partial V_j}{\partial V_k} = \delta_{ij} = \mathbb{E}\left[V_i V_j\right]$. Namely, when derivatives act in pairs they yield a Kronecker delta factor ($\delta_{ij}$), which is equal to the factor obtained from a covariance ($\mathbb{E}\left[V_i V_j\right]$). While Isserlis' theorem instructs us to sum over all possible covariance configurations (and therefore over all possible edge configurations), a pair of summed derivative leads to a particular covariance factor. Therefore, we should only consider graphs that include the edge corresponding to this covariance factor.

We are now ready to prove Theorem 1 for the case of single hidden layer with linear activations. A proof for the general case can be found in Appendix B.

*Proof (Theorem 1, one hidden layer).* Let $C(x_1, \ldots, x_m)$ be a correlation function for a network with a single hidden layer and linear activation. Let $G_C(V, E)$ be the cluster graph of $C$, and let $\gamma(V, E') \in \Gamma(C)$ be a Feynman diagram. Notice that $E \subset E'$. Indeed, $E'$ contains an edge corresponding to each pair of contracted derivative tensors in $C$, and $E = \{(v_i, v_j) \mid (T(x_i), T(x_j)) \text{ contracted in } C\}$. In addition, notice that $\gamma$ only contains connected

---

[8]We are not aware of a physics application in which such derivatives are included in a Feynman diagram description of correlation functions. Therefore, to our knowledge our treatment of these derivatives is novel both in machine learning and in physics.

components (*i.e.* loops) with an even number of vertices, because every vertex has exactly one edge of each type. Therefore, $n_\gamma \leq n_e + \frac{n_o}{2}$, where $n_\gamma$ is the number of loops in $\gamma$, and $n_e$ ($n_o$) is the number of even (odd) components in $G_C$. The bound is saturated when every even component of $G_C$ belongs to a different component of $\gamma$, and pairs of odd components in $G_C$ belong to different components of $\gamma$. From Theorem 3, we have that $C = \mathcal{O}(n^s)$ where $s = \max_{\gamma \in \Gamma(C)} n_\gamma - \frac{m}{2} \leq n_e + \frac{n_o}{2} - \frac{m}{2}$. □

## 5 DISCUSSION

Ensemble averages of the network function and its derivatives are an important class of functions that often show up in the study of wide neural networks. Examples include the ensemble average of the train and test losses, the covariance of the network function, and the Neural Tangent Kernel (Jacot et al. (2018)). In this work we presented Conjecture 1, which allows one to derive the asymptotic behavior of such functions at large width.

For the case of deep linear networks, we presented a complete analytic understanding of the Conjecture based on Feynman diagrams. In addition, we presented empirical and analytic evidence showing that the Conjecture also holds for deep networks with non-linear activations, as well as for networks with non-Gaussian initialization. We found that the Conjecture holds in all cases we tested.

The basic tools presented in this work can be applied to many aspects of wide network research, greatly simplifying theoretical calculations. We presented several applications of our method to the asymptotic behavior of wide networks during stochastic gradient descent, and additional applications are presented in Appendix E. We were able to improve upon known results by tightening existing bounds, and by applying the technique to SGD as well as to gradient flow. In addition, we took a step beyond the infinite width limit, deriving closed-form expressions for the first finite-width correction to the network evolution. These novel results open the door to studying finite-width networks by systematically expanding around the infinite width limit.

A central question in the study of wide networks is whether the infinite width limit is a good model for describing the behavior of realistic deep networks (Chizat et al. (2018); Ghorbani et al. (2019); Geiger et al. (2019b)). In this work we take a step toward answering this question, by working out the next order in a perturbative expansion around the infinite width limit, potentially bringing us closer to an analytic description of finite-width networks. We hope that the techniques presented here provide a basis to systematically answering these and other questions about the behavior of wide networks.

## ACKNOWLEDGEMENTS

The authors would like to thank Alex Alemi, Yasaman Bahri, Boris Hanin, Jared Kaplan, Jaehoon Lee, Behanm Neyshabur, Sam Schoenholz, Sylvia Smullin, and Jascha Sohl-Dickstein for useful discussion. The authors would especially like to thank Ying Xiao for extensive comments on early versions of this manuscript.

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

## A   EXPERIMENTAL DETAILS AND ADDITIONAL RESULTS

### A.1   NON-GAUSSIAN INITIALIZATION

In Table 1 we listed asymptotic bounds on several correlation functions, where the model parameters were initialized from a Gaussian distribution. Table 2 shows additional results using the same experimental setup, but with weights sampled uniformly from $\{\pm1\}$. We again find good agreement with the predictions of Conjecture 1.

| Correlation function $C$ | $n_e, n_o$ | $s_C$ | lin. | ReLU | tanh |
|---|---|---|---|---|---|
| $\mathbb{E}_\theta\left[f(x_1)f(x_2)\right]$ | 0,2 | 0 | 0.00 | 0.00 | 0.02 |
| $\mathbb{E}_\theta\left[f(x_1)f(x_2)f(x_3)f(x_4)\right]$ | 0,4 | 0 | -0.07 | 0.06 | -0.02 |
| $\sum_\mu \mathbb{E}_\theta\left[\partial_\mu f(x_1)\partial_\mu f(x_2)\right]$ | 1,0 | 0 | 0.00 | 0.00 | 0.00 |
| $\sum_{\mu,\nu} \mathbb{E}_\theta\left[\partial_\mu f(x_1)\partial_\nu f(x_2)\partial_{\mu,\nu} f(x_3)f(x_4)\right]$ | 0,2 | -1 | -1.02 | -1.01 | -0.97 |
| $\sum_{\mu,\nu,\rho} \mathbb{E}_\theta\left[\partial_\mu f(x_1)\partial_\nu f(x_2)\partial_\rho f(x_3)\partial_{\mu,\nu,\rho} f(x_4)\right]$ | 1,0 | -1 | -2.00 | -1.99 | -2.02 |
| $\sum_{\mu,\nu,\rho,\sigma} \mathbb{E}_\theta\left[\partial_\mu f(x_1)\partial_\nu f(x_2)\partial_{\mu,\nu} f(x_3)\partial_\rho f(x_4)\partial_\sigma f(x_5)\partial_{\rho,\sigma} f(x_6)\right]$ | 0,2 | -2 | -2.05 | -2.01 | -1.99 |

Table 2: Examples of bounds on correlation functions obtained from Conjecture 1. The experimental setup is the same as in Table 1, but the model parameters are sampled uniformly from $\{\pm 1\}$ instead of from a Gaussian distribution. We find good agreement with the theoretical predictoins, and in many cases the bound is tight.

## A.2 EXPERIMENTAL DETAILS

The experiments in Figure 1 were performed on two-class MNIST, computing a single randomly-chosen component of the kernel $\Theta$. Sub-figure (a) uses networks trained for 1024 steps with learning rate 1.0 and 1000 samples per class, averaged over 100 initializations. Each curve in figure (b) represents a single instance of the network map evaluated on a random image over the corse of training. The models were trained with 10 samples per class and learning rate 0.1. The input to the network is normalized by the square root of the input dimension as in (Jacot et al. (2018))

$$f(x) = n^{-1/2}V^T\sigma(n^{-1/2}W^{d-1}\cdots\sigma(n^{-1/2}W^1\sigma(D_{\text{in}}^{-1/2}Ux)))\,. \tag{15}$$

## B FEYNMAN DIAGRAMS FOR DEEP LINEAR NETWORKS

Feynman diagrams can be used to derive asymptotic upper bounds on deep linear networks in the large width limit. In this section we describe the method in detail, and use it to prove Theorem 1.

### B.1 FEYNMAN DIAGRAMS AND DOUBLE-LINE DIAGRAMS

In this section we build on the results of Section 4 and consider correlation functions of deep linear networks with $d$ hidden layers. The network function was defined in (2), and here we set the activation $\sigma$ to be the identity. Definition 2 describes how to map a correlation function $C$ to $\Gamma(C)$, a family of graphs called Feynman diagrams. The Feynman diagram method relies on Isserlis' theorem, which allows us to express arbitrary moments of multivariate Gaussian variables in terms of their covariance.

**Theorem 4** (Isserlis). *Let $z = (z_1, \ldots, z_l)$ be a centered multivariate Gaussian variable. For any positive integer $k$,*

$$\mathbb{E}_z\left[z_{i_1}\cdots z_{i_{2k}}\right] = \frac{1}{2^k k!}\sum_{\pi\in S_{2k}}\mathbb{E}\left[z_{i_{\pi(1)}}z_{i_{\pi(2)}}\right]\mathbb{E}\left[z_{i_{\pi(3)}}z_{i_{\pi(4)}}\right]\cdots\mathbb{E}\left[z_{i_{\pi(2k-1)}}z_{i_{\pi(2k)}}\right]\,, \tag{16}$$

$$\mathbb{E}_z\left[z_{i_1}\cdots z_{i_{2k-1}}\right] = 0\,. \tag{17}$$

*In particular, if the covariance matrix of $z$ is the identity then*

$$\mathbb{E}_z\left[z_{i_1}\cdots z_{i_{2k}}\right] = \frac{1}{2^k k!}\sum_{\pi\in S_{2k}}\delta_{i_{\pi(1)}i_{\pi(2)}}\delta_{i_{\pi(3)}i_{\pi(4)}}\cdots\delta_{i_{\pi(2k-1)}i_{\pi(2k)}}\,. \tag{18}$$

Using this theorem, a correlation function $C$ can be expressed as a sum over permutations as in (16). Each term in this sum maps to a Feynman diagram in $\Gamma(C)$.

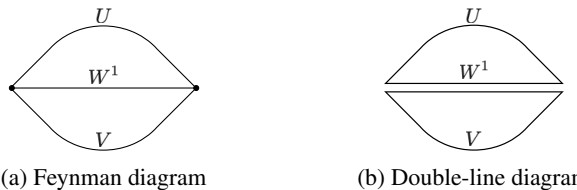

(a) Feynman diagram          (b) Double-line diagram

Figure 4: Feynman diagrams for $\mathbb{E}_\theta\left[f(x)f(x')\right]$ with 2 hidden layers. Notice that the $U, V$ edges in the Feynman diagram map to single edges in the double-line diagram, while the $W$ edge maps to a double edge.

As an example, consider the correlation function $C(x, x') = \mathbb{E}_\theta\left[f(x)f(x')\right]$ for a network with 2 hidden layers. An explicit calculation gives

$$C(x, x') = \frac{1}{n^2} \sum_{i,j,k,l}^{n} \sum_{\alpha,\beta}^{D_{\mathrm{in}}} \mathbb{E}_\theta\left[V_i W_{ij}^1 U_{j\alpha} V_k W_{kl}^1 U_{l\beta}\right] x_\alpha x'_\beta \tag{19}$$

$$= \frac{1}{n^2} \sum_{i,j,k,l}^{n} \sum_{\alpha,\beta}^{D_{\mathrm{in}}} \mathbb{E}_V\left[V_i V_k\right] \mathbb{E}_W\left[W_{ij}^1 W_{kl}^1\right] \mathbb{E}_U\left[U_{j\alpha}\, U_{l\beta}\right] x_\alpha x'_\beta \tag{20}$$

$$= \frac{x^T x'}{n^2} \sum_{i,k=1}^{n} \delta_{ik}\delta_{ik} \sum_{j,l=1}^{n} \delta_{jl}\delta_{jl} = x^T x' = \mathcal{O}(n^0)\,. \tag{21}$$

To get from (19) to (20), we applied Isserlis' theorem for every choice of indices $i, j, k, l, \alpha, \beta$. We find that there is at most one permutation $\pi$ of the network parameters such that the covariances do not vanish. This is because the covariance of parameters across different layers vanishes identically (for example $\mathbb{E}_\theta\left[V_i W_{jk}^1\right] = 0$). Correspondingly, this correlation function has a single Feynman diagram, shown in Figure 4a.

For networks with one hidden layer, we saw in Section 4 that the asymptotic behavior is determined by the number of loops in a graph. This observation does not immediately generalize to networks with general depth, because it is not obvious how to count loops in diagrams such as the one in Figure 4a. The problem can be traced back to the fact that weight matrices have covariances of the form $\mathbb{E}\left[W_{ij}W_{kl}\right] = \delta_{ik}\delta_{jl}$ involving two Kronecker deltas, but the procedure described so far assumes that each covariance (and each edge in the graph) corresponds to a single Kronecker delta.

A similar question appeared in the context of theoretical physics, and the correct generalization is due to 't Hooft (1974). The idea is to treat each Feynman diagram as the triangulation of a Riemann surface, and to define the number of loops in a graph to be the number of faces of the triangulation. In practice, this involves mapping each Feynman diagram to a new *double-line diagram*: A graph in which each edge corresponds to a single Kronecker delta factor, and loops correspond to triangulation faces of the original diagram.

**Definition 3.** *Let $\gamma \in \Gamma(C)$ be a Feynman diagram for a correlation function $C$ involving $k$ derivative tensors for a network of depth $d$. Its* double-line *graph, $\mathrm{DL}(\gamma)$ is a graph with $kd$ vertices of degree 2, defined by the following blow-up procedure.*

- *Each vertex $v^{(i)}$ in $\gamma$ is mapped to $d$ vertices $v_1^{(i)}, \ldots, v_d^{(i)}$ in $\mathrm{DL}(\gamma)$.*

- *Each edge $(v^{(i)}, v^{(j)})$ in $\gamma$ of type $U$ is mapped to a single edge $(v_1^{(i)}, v_1^{(j)})$.*

- *Each edge $(v^{(i)}, v^{(j)})$ in $\gamma$ of type $W^l$ is mapped to two edges $(v_l^{(i)}, v_l^{(j)})$, $(v_{l+1}^{(i)}, v_{l+1}^{(j)})$.*

- *Each edge $(v^{(i)}, v^{(j)})$ in $\gamma$ of type $V$ is mapped to a single edge $(v_d^{(i)}, v_d^{(j)})$.*

*The number of* faces *in $\gamma$ is given by the number of loops in the double-line graph $\mathrm{DL}(\gamma)$.*

Figure 4 shows the Feynman diagram and corresponding double-line diagram for $\mathbb{E}_\theta\left[f(x)f(x')\right]$ with 2 hidden layers. We can interpret this Feynman diagram as a triangulation of the disc: a 2-dimensional

surface with a single boundary. The triangulation has 2 vertices, 3 edges corresponding to the edges of the Feynman diagram, and 2 faces correponding to the loops of the double-line diagram. Figure 5 shows additional examples of double-line diagrams, and Figure 6 shows the double-line diagrams of a correlation function with derivatives. As explained in Section 4, contracted derivative tensors in a correlation function $C$ map to forced edges in $\Gamma(C)$, and these are marked with dashed lines on the diagrams.

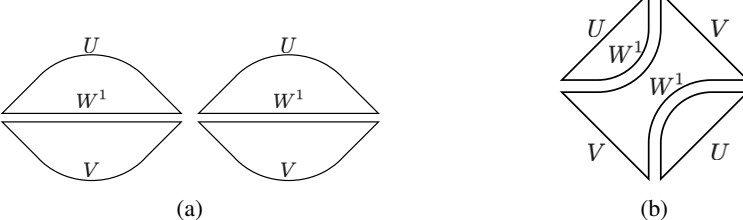

(a) (b)

Figure 5: Double-line diagrams for $\mathbb{E}_\theta \left[ f(x_1)f(x_2)f(x_3)f(x_4) \right]$ for a deep linear network with 2 hidden layers.

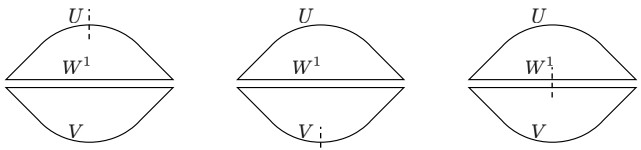

Figure 6: Double-line graphs describing the expectation value of the NTK, $\mathbb{E}_\theta \left[ \Theta \right]$, for a deep linear network with two hidden layers. Crossed edges mark the edges that are forced by contracted derivatives. The derivative can act on either $U$, $V$, or $W^1$, and therefore there are three diagrams. Each diagram has 2 vertices and 2 faces, and therefore the correlation function is $\mathcal{O}(n^0)$ according to the Feynman rules.

Generally, every Feynman diagram of a deep linear network can be interpreted as the triangulation of a 2-dimensional manifold with at least one boundary. Intuitively, the presence of the boundary is due to the fact that the $U, V$ weights at both ends of network have only one dimension that scales linearly with the width. As a result, in the double-line diagrams the $U, V$ edges become single lines rather than double lines. These 'missing' lines translate into fewer faces in the triangulation, and the 'missing' faces can be geometrically interpreted as boundaries in the corresponding surface.[9]

The discussion above is summarized by the following result, due to 't Hooft (1974), that describes how the asymptotic behavior of a general correlation function can be computed using the Feynman rules for deep linear networks.

**Theorem 5.** *Let $C(x_1, \ldots, x_m)$ be a correlation function of a deep linear network with $d$ hidden layers, and let $\gamma \in \Gamma(C)$ be a Feynman diagram. The diagram represents a subset of terms that contribute to $C$, and its asymptotic behavior is determined by the Feynman rules: the subset is $\mathcal{O}(n^{s_\gamma})$ where $s_\gamma = l_\gamma - \frac{dm}{2}$, and $l_\gamma$ is the number of loops in the double-line diagram $\mathrm{DL}(\gamma)$. Furthremore, the correlation function is $C = \mathcal{O}(n^s)$, where $s = \max_{\gamma \in \Gamma(C)} s_\gamma$.*

The intuition for the formula $s_\gamma = l_\gamma - \frac{dm}{2}$ is similar to the single hidden layer case, Theorem 3. The term $l_\gamma$ counts the number of factors of the form $\sum_{i_1, \ldots, i_k} \delta_{i_1 i_2} \delta_{i_2 i_3} \cdots \delta_{i_k i_1} = n$ that appear in the Correlation function after applying Isserlis' theorem. The term $\left( -\frac{dm}{2} \right)$ is due to the explicit $n^{-d/2}$ normalization of the network function.

---

[9]One can consider the *cyclic model* $\tilde{f}(x) = n^{-(d+1)/2} \mathrm{Tr} \left( \tilde{V} W^{d-1} \cdots W^1 \tilde{U} \right) x$ with 1D input $x$, in which all the weight tensors $\tilde{U}, \tilde{V}, W^1, \ldots, W^{d-1}$ are $n \times n$ matrices. The Feynman diagram construction for this model is similar to the deep linear case, except that all edges in the Feynman diagram map to double edges in the double-line diagrams. Such diagrams can be interpreted as triangulations of surfaces with no boundaries. Unlike the deep network diagrams, they have no 'missing' loops.

### B.2 ASYMPTOTICS OF DEEP LINEAR NETWORKS

We now prove Theorem 1. The theorem follows from the following lemma, again due to 't Hooft (1974), that relates the asymptotic behavior to the number of connected components in a Feynman diagram.

**Lemma 2.** *Let $C(x_1, \ldots, x_m)$ be a correlation function for a deep linear network. Let $c_\gamma$ be the number of connected components of a graph $\gamma \in \Gamma(C)$. Then $C = \mathcal{O}(n^s)$, where*

$$s = \max_{\gamma \in \Gamma(C)} c_\gamma - \frac{m}{2}. \tag{22}$$

*Proof.* We prove the result for 1D inputs ($D_{\text{in}} = 1$), and it is easy to generalize to arbitrary input dimension. Let $\gamma \in \Gamma(C)$ be a Feynman diagram and let $\gamma'$ be a connected component of $\gamma$ with $v_{\gamma'}$ vertices. Notice that we can apply the Feynman rules of Theorem 5 separately to each component $\gamma'$ and find a bound $\mathcal{O}(n^{s_{\gamma'}})$. Then, $\gamma = \mathcal{O}(n^{s_\gamma})$ where $s_\gamma = \sum_{\gamma'} s_{\gamma'}$, and the sum runs over the connected components of $s_\gamma$. We will show below that $s_{\gamma'} \leq 1 - \frac{v_{\gamma'}}{2}$, and therefore $s_\gamma \leq c_\gamma - \frac{m}{2}$, which is sufficient to prove (22).

Let us prove the remaining statement about $s_{\gamma'}$. The Euler character of the graph $\gamma'$ with $v = v_{\gamma'}$ vertices, $e$ edges and $f$ faces is $\chi = v - e + f$. The degree of each vertex in the graph is $d + 1$, and therefore $e = \frac{(d+1)v}{2}$. Using Theorem 5 the graph is $\mathcal{O}(n^{s_{\gamma'}})$ where $s_{\gamma'} = f - \frac{dv}{2}$. We therefore find that $\chi = \frac{v}{2} + s_{\gamma'}$. The graph $\gamma'$ is a triangulation of some connected surface with at least one boundary. The Euler character for such a surface is bounded by $\chi \leq 1$, and therefore $s_{\gamma'} \leq 1 - \frac{v}{2}$. $\square$

Let us now prove Theorem 1.

*Proof.* Let $C(x_1, \ldots, x_m)$ be a correlation function for a deep linear network. Suppose that the cluster graph $G_C$ has $n_e$ even size components and $n_o$ odd size components. Let $\gamma \in \Gamma(C)$ be a Feynman diagram with $c_\gamma$ connected components. We will show that $c_\gamma \leq n_e + \frac{n_o}{2}$. It then follows immediately from Lemma 2 that $C = \mathcal{O}(n^s)$ where $s = n_e + \frac{n_o}{2} - \frac{m}{2}$, concluding the proof.

Let us derive the bound on $c_\gamma$. First, all vertices that belong to a given cluster (a component of $G_C$) will also belong to the same connected component in $\gamma$. This is because every edge in $G_C$ is also an edge in $\gamma$ (note that $G_C$ and $\gamma$ have the same set of vertices). Therefore, $c_\gamma \leq n_e + n_o$. Second, note that every connected component of the graph $\gamma$ has an even number of vertices. Indeed, each edge has a type $t$, and each vertex has exactly one edge of each type. Therefore, a connected component with $v$ vertices has $\frac{v}{2}$ edges of each type, and so $v$ must be even. It follows that the vertices of even clusters can form their own connected components in a Feynman diagrams, while odd clusters must be connected in sets of 2 or more to form connected components. The bound on $c_\gamma$ then follows.

For deep linear networks with bias the proof is the same, except we use Lemma 3 below instead of Lemma 2. $\square$

### B.3 DEEP LINEAR NETWORKS WITH BIAS

In this section we argue that Conjecture 1 holds for deep linear networks with bias. Let us add bias to the definition (2) of a deep linear network (setting $\sigma$ to be the identity). We choose $x = 1$ without loss of generality. The activations $z^l$ at layer $l$ and the network function $f$ are given by

$$z^1 = U, \tag{23}$$

$$z^{l+1} = n^{-1/2} W^l z^l + b^l, \quad l = 1, \ldots, d-1, \tag{24}$$

$$f = n^{-1/2} V^T z^d. \tag{25}$$

Here $b^1, \ldots, b^{d-1} \in \mathbb{R}^n$ are biases whose elements are chosen independently from $\mathcal{N}(0, 1)$ at initialization. The network function can be written as

$$f = \sum_{k=2}^{d+1} n^{-(k-1)/2} V^T W^{d-1} \cdots W^{d+2-k} b^{d+1-k}, \tag{26}$$

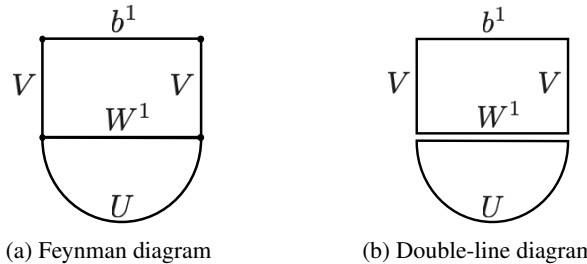

(a) Feynman diagram          (b) Double-line diagram

Figure 7: New Feynman diagrams for $\mathbb{E}_\theta \left[ f(x_1)f(x_2)f(x_3)f(x_4) \right]$ with 2 hidden layers and bias.

where we defined $b^0 = U$. In the rest of this Section we extend our definitions of Feynman diagrams and double-line diagrams to deep linear networks with bias, and then extend several of our results to this case.

**Definition 4.** *Let $C(x_1, \ldots, x_m)$ be a correlation function for a network with $d$ hidden layers and bias. The family $\Gamma(C)$ is the set of all graphs that have the following properties.*

1. *There are $m$ vertices $v_1, \ldots, v_m$, where each $v_i$ has degree $k_i \in \{2, \ldots, d+1\}$.*

2. *Each edge has a type $t$. Every vertex of degree $k$ has one edge of each type $t \in \{b^{d+1-k}, W^{d+2-k}, \ldots, W^{d-1}, V\}$.*

3. *If two derivative tensors $T_{\mu_1, \ldots, \mu_\ell}(x_i), T_{\nu_1, \ldots, \nu_{\ell'}}(x_j)$ are contracted $s$ times in $C$, the graph must have at least $s$ edges (of any type) connecting the vertices $v_i, v_j$.*

The intuition behind this definition is that term $k$ in the sum in (26) is equal to the network function of a deep linear network with depth $k-1$. Correlation functions of such a function can be represented diagramatically by including vertices of degree $k$. To represent the entire sum we must consider all diagrams with vertices of varying degree.

We now extend Definition 3 of double-line graphs to accommodate vertices with varying degree.

**Definition 5.** *Let $\gamma \in \Gamma(C)$ be a Feynman diagram for a correlation function $C$ involving $m$ derivative tensors for a network of depth $d$ with biases. Its double-line graph, $\mathrm{DL}(\gamma)$ is a graph with vertices of degree 2, defined by the following blow-up procedure.*

• *Each vertex $v^{(i)}$ in $\gamma$ of degree $k_i$ is mapped to $k_i - 1$ vertices $v_1^{(i)}, \ldots, v_{k_i-1}^{(i)}$ in $\mathrm{DL}(\gamma)$.*

• *Each edge $(v^{(i)}, v^{(j)})$ in $\gamma$ of type $W^l$ is mapped to two edges, $(v_{l+k_i-d-1}^{(i)}, v_{l+k_j-d-1}^{(j)})$ and $(v_{l+k_i-d}^{(i)}, v_{l+k_j-d}^{(j)})$.*

• *Each edge $(v^{(i)}, v^{(j)})$ in $\gamma$ of type $b^{d+1-k_i}$ is mapped to a single edge $(v_1^{(i)}, v_1^{(j)})$ (notice that $k_i = k_j$).*

• *Each edge $(v^{(i)}, v^{(j)})$ in $\gamma$ of type $V$ is mapped to a single edge $(v_{k_i-1}^{(i)}, v_{k_j-1}^{(j)})$.*

*As in Definition 3, the number of* faces *in $\gamma$ is given by the number of loops in the double-line graph $\mathrm{DL}(\gamma)$.*

Figure 7 is an example of the new single line and double line diagrams that appear in the correlation function $\mathbb{E}_\theta \left[ f(x_1)f(x_2)f(x_3)f(x_4) \right]$ for a two-hidden-layer network with biases.

Let us now extend Theorem 5 (the Feynman rules) to accomodate bias. The only difference is in the explicit normalization. Previously, each vertex had degree $d+1$ and contributed a factor of $n^{-d/2}$ due to the explicit normalization of the function map. With biases, as a result of (26), a vertex with degree $k$ contributes a factor of $n^{(k-1)/2}$.

**Theorem 6.** *Let $C(x_1, \ldots, x_m)$ be a correlation function of a deep linear network with $d$ hidden layers and with biases. Let $\gamma \in \Gamma(C)$ be a Feynman diagram. The diagram represents a subset of*

*terms that contribute to $C$, and its asymptotic behavior is determined by the Feynman rules: the subset is $\mathcal{O}(n^{s_\gamma})$ where $s_\gamma = l_\gamma - \sum_{i=1}^m \frac{k_i - 1}{2}$, with $k_i$ the degree of the $i$-th vertex, and $l_\gamma$ the number of loops in the double-line diagram $\mathrm{DL}(\gamma)$. Furthremore, the correlation function is $C = \mathcal{O}(n^s)$, where $s = \max_{\gamma \in \Gamma(C)} s_\gamma$.*

We now introduce the following result, which generalizes Lemma 2 to the case of networks with bias. This Lemma is used above in the proof of Theorem 1.

**Lemma 3.** *Let $C(x_1, \ldots, x_m)$ be a correlation function for a deep linear network with biases. Let $c_\gamma$ be the number of connected components of a graph $\gamma \in \Gamma(C)$. Then $C = \mathcal{O}(n^s)$, where*

$$s = \max_{\gamma \in \Gamma(C)} c_\gamma - \frac{m}{2}. \tag{27}$$

*Proof.* It is enough to show that each connected component $\gamma'$ in $\gamma$ is bounded as $\mathcal{O}(n^{s_{\gamma'}})$ where $s_{\gamma'} \leq 1 - \frac{v_{\gamma'}}{2}$. The graph $\gamma'$ is a triangulation of a Riemann surface with $f$ faces $e$ edges, and $v_{\gamma'}$ vertices of degrees $k_1, \ldots, k_{v_{\gamma'}}$. The Feynman rules give

$$s_{\gamma'} = f - \sum_{i=1}^{v_{\gamma'}} \frac{k_i - 1}{2}. \tag{28}$$

Using the relation $e = \sum_{i=1}^{v_{\gamma'}} \frac{k_i}{2}$ and the definition of the Euler character, $\chi = v_{\gamma'} - e + f$ we have

$$s_{\gamma'} = \chi - \frac{v_{\gamma'}}{2}. \tag{29}$$

The diagram $\gamma'$ is a triangulation of a Riemann surface with at least one boundary, thus $\chi \leq 1$ and $s_{\gamma'} \leq 1 - \frac{v_{\gamma'}}{2}$. □

## C NON-LINEARITIES

In previous sections we presented the Feynman diagram method for computing the large width asymptotics of correlation functions. In this section we show that the method applies as-is for deep networks with ReLU non-linearities and all-equal inputs, as well as to networks with a single hidden layer, a broader class of non-linearities, and arbitrary inputs. Theorem 2 follows immediately from the results presented in this section.

### C.1 DEEP NETWORKS WITH RELU NON-LINEARITIES

The following result guarantees that the presence of ReLU non-linearities does not change the asymptotic upper bound compared with linear activations, when all inputs are the same.

**Theorem 7.** *Let $f_{\mathrm{nl}}$ be a network function of the form (2) with ReLU activation. Let $f$ be a network with the same architecture but with linear activation. Let $C(x_1, \ldots, x_m; f_{\mathrm{nl}})$ be a correlation function of the deep network $f_{\mathrm{nl}}$ with width $n$, and let $f$ be a deep linear network with the same width and depth. Suppose that all inputs are the same, $x_1 = x_2 = \cdots = x_m$. Then*

$$C(x_1, \ldots, x_m; f_{\mathrm{nl}}) = \mathcal{O}(C(x_1, \ldots, x_m; f)). \tag{30}$$

Intuitively, we will rely on the fact that for ReLU networks we can, in some cases, treat the binary neuron activations as being statistically independent of the weights. This result is due to (Hanin & Nica (2018)). Given this result, we can bound the contribution of the binary activations, and the remaining Gaussian integral is equivalent to that found in a deep linear network. The proof does not work for correlation functions with non-equal inputs, because in that case the independence result of Hanin & Nica (2018) no longer holds.

*Proof.* We can write the network function as

$$f_{\mathrm{nl}}(x) = n^{-d/2} V^T D^d(x) W^{d-1} D^{d-1}(x) W^{d-2} \cdots D^2(x) W^1 D^1(x) U x, \tag{31}$$

$$D^j(x) := H(W^{j-1} \sigma(W^{j-2} \cdots \sigma(Ux))), \quad j = 1, \ldots, d. \tag{32}$$

Here, $H$ is the Heaviside step function acting elementwise on its vector argument. We now introduce the construction from Hanin & Nica (2018). Let $\xi^j, \eta^j,\ j = 1, \ldots, d$ be diagonal matrices of dimension $n$, whose diagonal elements are $\pm 1$-Bernoulli($p$) variables with $p = \frac{1}{2}$. We define the new variables

$$\hat{U} := \xi^1 U, \quad \hat{V} := \eta^d V, \quad \hat{W}^j := \xi^{j+1} W^j \eta^j, \quad j = 1, \ldots, d. \tag{33}$$

Let $\hat{f}_{\mathrm{nl}}$ be a network function with the same architecture as $f_{\mathrm{nl}}$ but using the re-defined weights. We define

$$\hat{f}_{\mathrm{nl}}(x) = n^{-d/2} \hat{V}^T \hat{D}^d(x) \hat{W}^{d-1} \cdots \hat{D}^2(x) \hat{W}^1 \hat{D}^1(x) \hat{U} x \tag{34}$$

$$= n^{-d/2} V^T \rho^d \hat{D}^d(x) W^{d-1} \rho^{d-1} \cdots \rho^2 \hat{D}^2(x) W^1 \rho^1 \hat{D}^1(x) U x, \tag{35}$$

$$\hat{D}^j(x) := H(\hat{W}^{j-1} \sigma(\hat{W}^{j-2} \cdots \sigma(\hat{U} x))), \quad j = 1, \ldots, d, \tag{36}$$

$$\rho^j := \eta^j \xi^j, \quad j = 1, \ldots, d. \tag{37}$$

The following was shown in Hanin & Nica (2018).

1. $f_{\mathrm{nl}}$ and $\hat{f}_{\mathrm{nl}}$ are equal in distribution.

2. $\{\hat{D}^j(x), \rho^j,\ j = 1, \ldots, d\}$ are independent of $\{U, V, W^1, \ldots, W^{d-1}\}$.

3. $\{\hat{D}^j(x),\ j = 1, \ldots, d\}$ are independent of each other for fixed $x$. The diagonal entries of each diagonal matrix $\hat{D}^j(x)$ are independent, and take the values $\{0, 1\}$ with probability $\frac{1}{2}$.

Now, the correlation function can be written as

$$C(x_1, \ldots, x_m; f_{\mathrm{nl}}) = \mathbb{E}_\theta \left[ F(x_1, \ldots, x_m; f_{\mathrm{nl}}) \right] = \mathbb{E}_{\theta, \eta, \xi} \left[ F(x_1, \ldots, x_m; \hat{f}_{\mathrm{nl}}) \right]. \tag{38}$$

The second equality follows from the fact that $f_{\mathrm{nl}} \stackrel{d}{=} \hat{f}_{\mathrm{nl}}$. Let us assume for now that the correlation function has no contracted derivatives. The we have

$$C(x_1, \ldots, x_m; f_{\mathrm{nl}}) = \mathbb{E}_{\theta, \eta, \xi} \left[ \hat{f}_{\mathrm{nl}}(x_1) \cdots \hat{f}_{\mathrm{nl}}(x_m) \right] \tag{39}$$

$$= \frac{1}{n^{dm/2}} \sum_{\vec{i}} \sum_{\vec{\alpha}} \mathbb{E} \left[ \prod_{l=1}^m V_{i_{l,d}} \right] \left( \prod_{l'=1}^{d-1} \mathbb{E} \left[ \prod_{l''=1}^m W^{l'}_{i_{l'',l'+1}, i_{l'',l'}} \right] \right) \mathbb{E} \left[ \prod_{l=1}^m U_{i_{l,1}, \alpha_l} \right] E_{\vec{i}, \vec{\alpha}}. \tag{40}$$

Here, $\vec{i} = \{i_{1,1}, \ldots, i_{m,d}\}$ and $\vec{\alpha} = \{\alpha_1, \ldots, \alpha_m\}$, and

$$E_{\vec{i}, \vec{\alpha}} := \mathbb{E} \left[ \prod_{l=1}^d \prod_{l'=1}^m \rho^l_{i_{l',l}} \hat{D}^l_{i_{l',l}}(x_{l'}) \right] \prod_{l=1}^m x^l_{\alpha_l}. \tag{41}$$

In writing the equation (40) we used the facts that $\hat{D}, \rho$ are independent of the parameters, and that parameters in different layers are independent. We now use Theorem 4 (Isserlis), which says that each of the expectation values over products of $V$, $W^j$, and $U$ elements is equal to a sum over permutations, where each term is a product over Kronecker delta functions — the covariance matrices of the parameters.

$$C(x_1, \ldots, x_m; f_{\mathrm{nl}}) = \frac{1}{n^{dm/2}} \sum_{\sigma_1, \ldots, \sigma_{d+1} \in S_m} \sum_{\vec{i}}^n \sum_{\vec{\alpha}}^{D_{\mathrm{in}}} \tilde{\Delta}^{\vec{\sigma}}_{\vec{i}, \vec{\alpha}} E_{\vec{i}, \vec{\alpha}}. \tag{42}$$

Here, $\tilde{\Delta}^{\vec{\sigma}}_{\vec{i}, \vec{\alpha}}$ is a product of Kronecker delta functions. The precise form of this object will not be important for our purpose, though it can be easily derived using Theorem 4.

We can now bound the correlation function as follows.

$$|C(x_1, \ldots, x_m; f_{\mathrm{nl}})| \leq \frac{1}{n^{dm/2}} \sum_{\sigma_1, \ldots, \sigma_{d+1} \in S_m} \sum_{\vec{i}}^n \sum_{\vec{\alpha}}^{D_{\mathrm{in}}} \tilde{\Delta}^{\vec{\sigma}}_{\vec{i}, \vec{\alpha}} \left| E_{\vec{i}, \vec{\alpha}} \right| \tag{43}$$

$$\leq \frac{E_{\max}}{n^{dm/2}} \sum_{\sigma_1, \ldots, \sigma_{d+1} \in S_m} \sum_{\vec{i}}^n \sum_{\vec{\alpha}}^{D_{\mathrm{in}}} \tilde{\Delta}^{\vec{\sigma}}_{\vec{i}, \vec{\alpha}} \tag{44}$$

$$= E_{\max} |C(v, \ldots, v; f)|. \tag{45}$$

Here, $E_{\max} = \max_{\vec{i}, \vec{\alpha}} \left| E_{\vec{i}, \vec{\alpha}} \right|$, and $v \in \mathbb{R}^{D_{\text{in}}}$ is the all-ones vector. The diagonal elements of $\rho^l, \hat{D}^l(x)$ are identical variables. Therefore, $E_{\vec{i}, \vec{\alpha}}$ only takes an $\mathcal{O}(1)$ number of values in the large width limit. Note that each of these values are independent of $n$, and therefore $E_{\max} = \mathcal{O}(1)$. We managed to bound the correlation function for $f_{\text{nl}}$ in terms of the correlation function for the corresponding linear network $f$ with fixed inputs. The asymptotics of the linear-network correlation function do not depend on the inputs, and therefore this concludes the proof. □

## C.2 SINGLE HIDDEN LAYER NETWORKS

For networks with a single hidden layer, defined by

$$f_{\text{nl}}(x) = \frac{1}{\sqrt{n}} \sum_{i=1}^{n} V_i \sigma(U_i^T x), \tag{46}$$

we can extend our asymptotic analysis to smooth non-linearities. We will show in Theorem 8 that for any correlation function $C$, we have

$$C(x_1, \ldots, x_m; f_{\text{nl}}) = \mathcal{O}(C(x_1, \ldots, x_m; f)), \tag{47}$$

where $f$ is a deep linear network of equal width and sufficient depth. Therefore, computing the asymptotics using Feynman diagrams for deep linear networks yields a bound on networks with a single hidden layer and smooth non-linearities.

Before delving into the proof of this claim, we consider a few simple examples. Let us begin with the correlation function $\mathbb{E}_\theta \left[ f_{\text{nl}}(x) f_{\text{nl}}(x') \right]$.

$$\mathbb{E}_\theta \left[ f_{\text{nl}}(x) f_{\text{nl}}(x') \right] = \frac{1}{n} \sum_{i,j}^{n} \mathbb{E}_\theta \left[ V_i V_j \sigma(U_i^T x) \sigma(U_j^T x') \right] \tag{48}$$

$$= \frac{1}{n} \sum_{i}^{n} \mathbb{E}_\theta \left[ \sigma(U_i^T x) \sigma(U_i^T x') \right] = \mathcal{O}(1). \tag{49}$$

Here we used two facts. First, the weights $V_i$ are unaffected by the non-linearity, so we can carry out the $V$ integral. Second, the summand in the last equation, $\mathbb{E} \left[ \sigma(U_i^T x) \sigma(U_i^T x') \right]$, is independent of both $i$ and $n$ because $U_i$ are i.i.d. variables.

Next, consider the following correlation function. For simplicity, here we set the input dimension to be $D_{\text{in}} = 1$, and we set all inputs to 1; this does not change the asymptotics.

$$\mathbb{E}_\theta \left[ \frac{d\Theta}{dt} \right] = \sum_{j,k}^{n} \mathbb{E}_\theta \left[ f_{\text{nl}} \frac{\partial^2 f_{\text{nl}}}{\partial U_j \partial V_k} \frac{\partial f_{\text{nl}}}{\partial U_j} \frac{\partial f_{\text{nl}}}{\partial V_k} \right]$$

$$= \frac{1}{n^2} \sum_{i_1, i_2}^{n} \mathbb{E}_\theta \left[ V_{i_1} V_{i_2} \sigma(U_{i_1}) \sigma'(U_{i_2}) \sigma'(U_{i_2}) \sigma(U_{i_2}) \right]$$

$$= \frac{1}{n^2} \sum_{i_1}^{n} \mathbb{E}_\theta \left[ \sigma(U_{i_1})^2 \sigma'(U_{i_1})^2 \right]$$

$$= \mathcal{O}(n^{-1}). \tag{50}$$

In the last line, we again used the fact that the summands are equal and independent of $n$.

In general, a correlation function $C(x_1, \ldots, x_m; f_{\text{nl}})$ can be reduced to the form

$$C = \frac{1}{n^{m/2}} \sum_{\alpha=1}^{K} \sum_{i_1, \ldots, i_{r_\alpha}}^{n} \mathcal{S}_{i_1 \ldots i_{r_\alpha}}^{(\alpha)}, \tag{51}$$

$$\mathcal{S}_{i_1 \ldots i_{r_\alpha}}^{(\alpha)} := \mathbb{E}_U \left[ \left( \sigma^{(\ell_1^\alpha)}(U_{i_1}^T x_{\bar{\sigma}_\alpha(1)}) \cdots \sigma^{(\ell_{k_1}^\alpha)}(U_{i_1}^T x_{\bar{\sigma}_\alpha(k_1^\alpha)}) \right) \times \cdots \times \right.$$
$$\left. \left( \sigma^{(\ell_{m+1-k_{r_\alpha}^\alpha}^\alpha)}(U_{i_{r_\alpha}}^T x_{\bar{\sigma}_\alpha(m+1-k_{r_\alpha}^\alpha)}) \cdots \sigma^{(\ell_m^\alpha)}(U_{i_{r_\alpha}}^T x_{\bar{\sigma}_\alpha(m)}) \right) F^{(\alpha)}(x_1, \ldots, x_m) \right].$$

$$\tag{52}$$

This form is obtained by carrying out the $V_i$ integrals, as well as all the sums that can be trivially carried out due to the presence of Kronecker deltas. Here, the $\alpha$ sum represents a sum over all $K$ terms that appear from performing the $V_i$ integrals using Isserlis' theorem; $\sigma^{(\ell)}$ denotes the $\ell$-th derivative of the non-linearity; $k_s^\alpha$ is the number of $\sigma^{(\cdot)}(U_{i_s}^T x)$ factors sharing the $i_s$ index; $\bar{\sigma}_\alpha \in S_m$ is a permutation; and $F^{(\alpha)}$ is a function whose exact form is not important for us. We will denote by $r_{\max}$ the maximum number of index sums appearing in (51), namely $r_{\max} := \max_\alpha r_\alpha$.

Our approach to establishing the asymptotic scaling will be to first bound the maximum number of index sums appearing in any term in our correlation function, written in the form (51), and then to argue that the summands are bounded by an $n$-independent constant.

Let us introduce a family of diagrams, $\Gamma'(C)$, which are different in general than the Feynman diagrams. A given diagram $g \in \Gamma'(C)$ is constructed as follows.

**Definition 6.** *Let $C(x_1, \ldots, x_m; f_{\mathrm{nl}})$ be a correlation function, where $f_{\mathrm{nl}}$ is the output of a network with one hidden layer, defined in (46). The family $\Gamma'(C)$ is the set of all graphs that have the following properties.*

- *Each derivative tensor in $C$ is mapped to a vertex in the graph.*

- *Each edge has a type that corresponds to one of the weight vectors $U$ or $V$.*

- *Each vertex has exactly one edge of $V$ type.*

- *If two derivative tensors are contracted in $C$, the graph must have at least one edge (of any type) connecting the corresponding vertices for each contraction.*

The reason for introducing this graphical structure is that it allows us to make two important statements. If we define $\tilde{c}_g$ the number of connected components in a graph $g \in \Gamma'(C)$ and $\tilde{c}_{\max} := \max_{g \in \Gamma'(C)} \tilde{c}_g$ then the following holds.

1. The maximal number of sums appearing in a correlation function of the form (51) is $\tilde{c}_{\max}$.

2. $\tilde{c}_{\max} = c_{\max}$, where $c_{\max}$ is the maximal number of connected components in the family of Feynman diagrams corresponding to a correlation function $C(x_1, \ldots, x_m; f_d)$, where $f_d$ is a deep linear network with $d$ hidden layers, and $d$ is large enough that none of the derivative tensors vanish.[10]

These two results, combined with a bound on the summands occurring in $C(x_1, \ldots, x_m; f_{\mathrm{nl}})$ will establish the bound (47).

**Lemma 4.** *A correlation function $C$ has $r_{max} = \tilde{c}_{max}$.*

*Proof.* Each vertex corresponds to a derivative tensor, $T_{\mu_1 \ldots \mu_k}$, which contains a single sum over paired $U_i$ and $V_i$ indices. If two vertices are connected by one or more edges, there is a Kronecker delta factor that sets the corresponding indices to be equal (due either to a derivative contraction, or to a $V$ covariance), reducing the number of sums by one. The result is that any connected component of a graph $g \in \Gamma'(C)$ corresponds to a single sum, and the total numer of index sums corresponding to a particular graph is $\tilde{c}_g$, the number of connected components. As a result, the maximum number of sums corresponding to the collection of graphs $\Gamma'(C)$ is $\tilde{c}_{\max}$. $\square$

**Lemma 5.** *The maximal number of connected components, $\tilde{c}_{max}$, over the collection of graphs $\Gamma'(C)$ corresponding to $C(x_1, \ldots, x_m; f_{nl})$ is bounded by the maximal number of components, $c_{max}$, over the collection $\Gamma(C)$ corresponding to $C(x_1, \ldots, x_m; f_d)$ of Feynman diagrams, where $f_d$ is a deep linear network of sufficient depth $d$.*

*Proof.* Let $n_e(n_o)$ be the number of even(odd) clusters in the cluster graph $G_C$ of $C(x_1, \ldots, x_m; f_d)$. The cluster graph, $G_C$ is a subgraph of any graph $g \in \Gamma'(C)$. We can thus think about the embedding of even and odd clusters into $g$. In any graph $g \in \Gamma'(C)$, an even cluster may belong to its own connected component, while for odd clusters there must be an even number of them in any connected

---

[10]Note that any derivative tensor of $f$ that has rank greater than $d$ vanishes.

component. This is because an even (odd) cluster contains an even (odd) number of factors of $V$, which must be paired up in any connected component. We find that

$$\tilde{c}_{\max} = n_e + \frac{n_o}{2} \leq c_{\max} . \tag{53}$$

The last inequality was used below Lemma 2 in the proof of Theorem 1. $\qquad\square$

**Theorem 8.** *Let $C(x_1, \ldots, x_m; f_{\mathrm{nl}})$ be a correlation function for a one hidden layer network. Let $c_{\max}$ be the maximal number of connected components over the collection of graphs $\Gamma(C)$ corresponding to $C(x_1, \ldots, x_m; f_d)$, where $f_d(x)$ is a deep linear network map, with with depth $d$ greater than or equal to the maximum number of derivatives appearing in any single derivative tensor in $C$. Then $C = \mathcal{O}(n^s)$ where $s = c_{\max} - \frac{m}{2}$. Furthermore,*

$$C(x_1, \ldots, x_m; f_{\mathrm{nl}}) = \mathcal{O}(C(x_1, \ldots, x_m; f)) . \tag{54}$$

*Proof.* The correlation function in (51) can be bound as

$$|C(x_1, \ldots, x_m; f_{\mathrm{nl}})| \leq \frac{1}{n^{m/2}} \sum_{\alpha=1}^{K} \sum_{i_1, \ldots, i_r}^{n} \left| \mathcal{S}^{(\alpha)}_{i_1, \ldots, i_{r_\alpha}} \right| . \tag{55}$$

For fixed inputs, $\mathcal{S}^{(\alpha)}_{i_1, \ldots, i_{r_\alpha}}$ can take at most $\mathcal{O}(1)$ different values as a function of its indices, and the values are independent of $n$. This is because the variables $U_i$ are identical. We define $s_{\max}$ as the maximum value of $|\mathcal{S}^{(\alpha)}_{i_1, \ldots, i_{r_\alpha}}|$ as a function of $\alpha$ and the indices. Combining this with the above lemmas we can then write.

$$|C(x_1, \ldots, x_m; f_{\mathrm{nl}})| \leq K s_{\max} n^{c_{\max} - m/2} . \tag{56}$$

The result of the theorem follows from Lemma 2. $\qquad\square$

# D CORRELATION FUNCTION ASYMPTOTICS

In this section we prove several general results about correlation function asymptotics in the large width limit. Throughout this section, we assume that Conjecture 1 holds.

## D.1 VARIANCE ASYMPTOTICS

Conjecture 1 can be used to bound the variance of the integrands that appear inside correlation functions.

**Lemma 6.** *Let $\tilde{C}(x_1, \ldots, x_m) = \mathbb{E}_\theta [F_\theta(x_1, \ldots, x_m)]$ be a correlation function, and let $G_C$ be the cluster graph of $C$ with $n_e$ even components and $n_o$ odd components. Assume Conjecture 1 holds such that $C = \mathcal{O}(n^{s_C})$, where $s_C = n_e + \frac{n_o}{2} - \frac{m}{2}$. Then $\mathrm{Var}_\theta [F_\theta(x_1, \ldots, x_m)] = \mathcal{O}(n^{2s_C})$.*

*Proof.* To bound the variance, it is enough to bound the correlation function

$$\tilde{C}(x_1, \ldots, x_{2m}) := \mathbb{E}_\theta [F_\theta(x_1, \ldots, x_m) F_\theta(x_{m+1}, \ldots, x_{2m})] , \tag{57}$$

because $\mathrm{Var}_\theta [F_\theta(x_1, \ldots, x_m)] \leq \tilde{C}(x_1, \ldots, x_m, x_1, \ldots, x_m)$. The correlation function $\tilde{C}$ has $2n_e$ even clusters and $2n_o$ odd clusters. It follows from Conjecture 1 that $\tilde{C} = \mathcal{O}(n^{2s_C})$. $\qquad\square$

As a corrolary, notice that if $C = \mathcal{O}(n^s)$ according to Conjecture 1, then typical realizations of the integrand will also be $\mathcal{O}(n^s)$. In other words, the asymptotic bound of Conjecture 1 holds for typical initializations, not just in expectation.

Table 3 shows empirical results for the variance of several functions. In all cases we find that the bound of Lemma 6 holds. For $\sum_\mu \mathrm{Var}_\theta [\partial_\mu f(x_1) \partial_\mu f(x_2)]$ we prove a tight bound below in Appendix E.1 for deep linear networks.

| Function | $n_e, n_o$ | $2s_C$ | lin. | ReLU | tanh |
|---|---|---|---|---|---|
| $\text{Var}_\theta\left[f(x_1)f(x_2)\right]$ | 0,2 | 0 | -0.08 | -0.00 | -0.02 |
| $\text{Var}_\theta\left[f(x_1)f(x_2)f(x_3)f(x_4)\right]$ | 0,4 | 0 | -0.03 | 0.02 | -0.05 |
| $\sum_\mu \text{Var}_\theta\left[\partial_\mu f(x_1)\partial_\mu f(x_2)\right]$ | 1,0 | 0 | -1.01 | -1.02 | -0.99 |
| $\sum_{\mu,\nu} \text{Var}_\theta\left[\partial_\mu f(x_1)\partial_\nu f(x_2)\partial_{\mu,\nu} f(x_3)f(x_4)\right]$ | 0,2 | -2 | -2.1 | -2.13 | -2.14 |
| $\sum_{\mu,\nu,\rho} \text{Var}_\theta\left[\partial_\mu f(x_1)\partial_\nu f(x_2)\partial_\rho f(x_3)\partial_{\mu,\nu,\rho} f(x_4)\right]$ | 1,0 | -2 | -4.02 | -4.1 | -3.05 |
| $\sum_{\mu,\nu,\rho,\sigma} \text{Var}_\theta\left[\partial_\mu f(x_1)\partial_\nu f(x_2)\partial_{\mu,\nu} f(x_3)\partial_\rho f(x_4)\partial_\sigma f(x_5)\partial_{\rho,\sigma} f(x_6)\right]$ | 0,2 | -4 | -4.09 | -4.14 | -4.01 |

Table 3: Bounds on variances obtained from Lemma 6, where the predicted exponent is $2s_C$, compared with empirical results. The predicted exponent is $2s_C$, and the 3 right-most columns list the empirical exponents. The experimental setup is the same as that of Table 1.

## D.2 GRADIENT FLOW

The following results are used in the gradient flow calculations of Section 3.

**Lemma 7.** *Let $G'$ be a graph with $m'$ vertices, $n'_e$ even components, and $n'_o$ odd components. Let $G$ be a subgraph of $G'$ with $m$ vertices, $n_e$ even components, and $n_o$ odd components. Then $s(n_e, n_o, m) \geq s(n'_e, n'_o, m')$ where $s(a,b,c) := a + \frac{b-c}{2}$.*

*Proof.* It is enough to show that $s(n_e, n_o, m)$ does not increase if we (1) add a vertex to $G$, or (2) add an edge to $G$, because $G'$ can be obtained from $G$ by performing such operations finitely many times. Adding a vertex to $G$ changes $n_e \mapsto n_e$, $n_o \mapsto n_o + 1$, and $m \mapsto m + 1$, leaving $s(n_e, n_o, m)$ unchanged. Next, if we add an edge to $G$ then $m$ does not change, and there are 4 possibilities for how $n_e$ and $n_o$ change.

1. The edge connects two even components. Then $n_e \mapsto n_e - 1$, $n_o \mapsto n_o$, and $s(n_e, n_o, m)$ decreases by 1.

2. The edge connects two odd components. Then $n_e \mapsto n_e + 1$, $n_o \mapsto n_o - 2$, and $s(n_e, n_o, m)$ does not change.

3. The edge connects an even component and an odd component. Then $n_e \mapsto n_e - 1$, $n_o \mapsto n_o$, and $s(n_e, n_o, m)$ decreases by 1.

4. The edge connects two vertices that belong to the same component. In this case $n_e$, $n_o$, and $s(n_e, n_o, m)$ do not change.

$\square$

We now prove Lemma 1 giving the scaling of time derivatives of correlation functions at initialization. We prove the result for polynomial loss functions. Extension to more general loss functions requires interchanging the large width limit and the Taylor expansion of the loss, which we do not discuss.

*proof (Lemma 1).* Notice that

$$\mathbb{E}_\theta\left[\frac{d^k F(x_1,\ldots,x_m)}{dt^k}\right] = \mathbb{E}_\theta\left[\left(\sum_\mu \sum_{(x',y')\in D_{\text{tr}}} \frac{\partial\ell(x',y')}{\partial f}\frac{\partial f(x')}{\partial\theta^\mu}\frac{\partial}{\partial\theta^\mu}\right)^k F(x_1,\ldots,x_m)\right].$$

(58)

On the right-hand side we have a linear combination $\sum_A \alpha_A C_A$ of correlation functions $C_A$, where the coefficients depend on the training set labels. Here we used the polynomial loss assumption. For each correlation function $C_A$, its integrand can be obtained from $F$ by finitely many operations of the form (1) multiply the integrand by $f(x)$ for some input $x$, and (2) act with a pair of contracted

derivatives on two of the derivative tensors. In the cluster graph representation, these two operations correspond to (1) adding a vertex, and (2) adding an edge. Therefore, the cluster graph $G_C$ of $C = \mathbb{E}_\theta[F]$ is a subgraph of the cluster graph $G_{C_A}$ of each one of the correlation functions $C_A$. By Lemma 7 we have that $C_A = \mathcal{O}(n^{s_C})$ for all $A$, and therefore the bound applies to $\mathbb{E}_\theta[d^k F/dt^k]$ as well. $\qquad\square$

### D.3 STOCHASTIC GRADIENT DESCENT

In this section we show that the asymptotics of a correlation function do not change under stochastic gradient descent (SGD) updates. Let $C(x_1, \ldots, x_m) = \mathbb{E}_{\theta \sim \mathcal{P}_0}[F_\theta(x_1, \ldots, x_m)]$ be a correlation function, where $F_\theta$ is the integrand which explicitly depends on the parameters $\theta$. Under a single SGD step, the parameters are updated as $\theta_{t+1} = \theta_t - \eta \nabla L_t(\theta_t)$, where $L$ is the mini-batch loss at time $t$. Let $\mathcal{P}_t$ denote the distribution of parameters at step $t$, where $\mathcal{P}_0$ is the initial distribution. We define the evolved correlation function at step $t$ to be

$$C_t(x_1, \ldots, x_m) := \mathbb{E}_{\theta \sim \mathcal{P}_t}[F_\theta(x_1, \ldots, x_m)] . \tag{59}$$

We have the following

**Theorem 9.** *Let $C$ be a correlation function for a network with linear activations, and assume that Conjecture 1 holds, namely $C = \mathcal{O}(n^{s_C})$ where $s_C$ is defined in the Conjecture. If $C_t$ is the evolved correlation function after $t$ SGD steps, then $C_t = \mathcal{O}(n^{s_C})$.*

*Proof.* Notice that

$$C_{t+1} = \mathbb{E}_{\theta \sim \mathcal{P}_{t+1}}[F_\theta] = \mathbb{E}_{\theta \sim \mathcal{P}_t}\left[F_{\theta - \eta \nabla L(\theta)}\right] . \tag{60}$$

The integrand $F_\theta$ can be written as a product of derivative tensors of the form $T_{\mu_1 \ldots \mu_k}(x; \theta)$, with contracted derivatives. Suppose that the network has $d$ hidden layers. Then under an SGD step, we have

$$T_{\mu_1 \ldots \mu_l}(x; \theta - \eta \nabla L_t(\theta)) = \sum_{k=0}^{d+1} \frac{(-\eta)^k}{k!} \sum_{\nu_1 \ldots \nu_k} \frac{\partial L_t}{\partial \theta^{\nu_1}} \cdots \frac{\partial L_t}{\partial \theta^{\nu_k}} T_{\mu_1 \ldots \mu_l \nu_1 \ldots \nu_k}(x; \theta) \tag{61}$$

$$= \sum_{k=0}^{d+1} \frac{(-\eta)^k}{k!} \sum_{x_1', \ldots, x_k' \in D_B} \left[ \frac{\partial L_t}{\partial f(x_1')} \cdots \frac{\partial L_t}{\partial f(x_k')} \right.$$

$$\left. \times \sum_{\nu_1, \ldots, \nu_k} T_{\nu_1}(x_1') \cdots T_{\nu_k}(x_k') T_{\mu_1 \ldots \mu_l \nu_1 \ldots \nu_k}(x) \right] . \tag{62}$$

Here $D_B$ is the mini-batch, and $x_a'$ are mini-batch samples. The $k$ sum is truncated because higher-order derivatives of $f$ vanish.

We can now see how taking a gradient descent step affects the cluster graph. After taking a step, each derivative tensor in $C_t$ is replaced by a sum (over $k, x_1', \ldots, x_k'$). Each term in the combination of these sums is a correlation function, whose cluster graph is a subgraph of $C$. Therefore, by Lemma 7, $C_t = \mathcal{O}(n^{s_C})$. $\qquad\square$

We note that for general activation functions the sum in (62) may be infinite. In this case, to complete the proof we would need to show that the infinite sum obeys the same bound as each individual term in the sum. We leave this for future work.

## E APPLICATIONS

Here we present several applications of our Feynman diagram method for computing large width asymptotics. We assume throughout this section that Conjecture 1 holds.

### E.1 NEURAL TANGENT KERNEL

In this section we prove two results regarding the NTK at large width. We show that the kernel converges in probability, and that during gradient descent it is constant up to $\mathcal{O}(n^{-1})$ corrections.

**Theorem 10.** *The Neural Tangent Kernel $\Theta$ of a deep linear network converges in probability in the large width limit, and its variance is $\mathcal{O}(n^{-1})$.*

Conjecture 1 is not sufficient for proving this theorem, as we need to use a more detailed Feynman diagram argument. Therefore, we only prove the theorem for the case of deep linear networks.

*Proof.* First, we will show that $\text{Var}[\Theta] = \mathcal{O}(n^{-1})$. Given a model function $f$, the variance is given by

$$\text{Var}[\Theta(x, x')] = A(x, x') - B(x, x'), \tag{63}$$

$$A(x, x') = \sum_{\mu, \nu} \mathbb{E}\left[\frac{\partial f^{(1)}(x)}{\partial \theta_\mu} \frac{\partial f^{(2)}(x')}{\partial \theta_\mu} \frac{\partial f^{(3)}(x)}{\partial \theta_\nu} \frac{\partial f^{(4)}(x')}{\partial \theta_\nu}\right], \tag{64}$$

$$B(x, x') = \sum_{\mu, \nu} \mathbb{E}\left[\frac{\partial f^{(1)}(x)}{\partial \theta_\mu} \frac{\partial f^{(2)}(x')}{\partial \theta_\mu}\right] \mathbb{E}\left[\frac{\partial f^{(3)}(x)}{\partial \theta_\nu} \frac{\partial f^{(4)}(x')}{\partial \theta_\nu}\right]. \tag{65}$$

Here $f^{(1)} = \cdots = f^{(4)} = f$; we introduced the superscripts so we can easily refer to individual factors. The crux of the argument is that the set of Feynman diagrams representing the expression (63) includes only connected graphs. Assuming this is the case, according to the bound (22) these diagrams will all scale as $\mathcal{O}(n^{(2-4)/2}) = \mathcal{O}(n^{-1})$ and so will the variance, completing the proof. To see why only connected graphs contribute, notice that

- $A(x, x')$ includes all diagrams in which the vertices corresponding to $f^{(1)}, f^{(2)}$ share an edge, and also the vertices $f^{(3)}, f^{(4)}$ share an edge (due to the explicit derivatives);

- $B(x, x')$ includes all diagrams in which *all* edges are either between $f^{(1)}, f^{(2)}$ or between $f^{(3)}, f^{(4)}$.

Therefore, in the full expression (63), the only remaining diagrams (*i.e.* the diagrams that do not cancel between the two terms) are those that include

- an edge connecting $f^{(1)}, f^{(2)}$,

- an edge connecting $f^{(3)}, f^{(4)}$, and

- an edge connecting one of $f^{(1)}, f^{(2)}$ to one of $f^{(3)}, f^{(4)}$.

These diagrams are all connected graphs, and this completes the proof that $\text{Var}[\Theta] = \mathcal{O}(n^{-1})$.

Here and above we have established that $\text{Var}[\Theta] = \mathcal{O}(n^{-1})$ and $\mathbb{E}[\Theta] = \mathcal{O}(n^0)$. More generally, let us consider a random variable $O$ equal to the product of $f$ and its derivatives, where the derivatives indices are fully summed in pairs. As we will now show, if $\mathbb{E}[O] = \mathcal{O}(n^0)$ and $\text{Var}[O] = \mathcal{O}(n^{-1})$ then (1) the limit $\lim_{n\to\infty} \mathbb{E}[O]$ exists, and (2) $O$ converges in probability to this limit. In particular, the NTK is an example of such a random variable, and so this will conclude the proof that the NTK converges in probability.

First, consider the mean. There is a finite number of diagrams contributing to $\mathbb{E}[O]$, and each has a well-defined $n$ scaling. Therefore, we can write

$$\mathbb{E}[O] = \sum_{k=0}^{k_{\max}} \frac{c_k}{n^k} \tag{66}$$

for some values of $k_{\max} \geq 0$ (integer) and $c_0, c_1, \ldots \in \mathbb{R}$. We see that the mean has a well-defined large $n$ limit,

$$\lim_{n\to\infty} \mathbb{E}[O] = c_0. \tag{67}$$

Next, let us show that $O - \mathbb{E}[O]$ converges in probability using Chebyshev's inequality. Indeed, by the variance assumption there exists $\tilde{c} > 0$ such that

$$P(|O - \mathbb{E}[O]| > \epsilon) \leq \frac{\text{Var}[O]}{\epsilon^2} \leq \frac{\tilde{c}}{n\epsilon^2} \to 0. \tag{68}$$

Combining the facts that $O - \mathbb{E}[O] \to_p 0$ and $\mathbb{E}[O] \to c_0$, we find that $O$ converges in probability to $c_0$. $\qquad \square$

Next, we show that the large width NTK is constant during training, and compute the asymptotics of the higher-order terms. The following argument is phrased for deep linear networks. More generally, the same argument holds for deep networks with smooth non-linear activations under the additional assumption that the large width limit and Taylor series can be exchanged (note that for such networks, the network function is analytic in the weights).

**Theorem 11.** *Let $f(x)$ be the network output of a deep linear network with MSE loss $L$. Let $\Theta_t(x, x')$ be the Neural Tangent Kernel at SGD step $t$, for some inputs $x, x'$. Then in the large width limit, the kernel is constant in $t$ in expectation, and*

$$\mathbb{E}_{\theta \sim \mathcal{P}_0}[\Theta_t(x, x') - \Theta_0(x, x')] = \mathcal{O}(n^{-1}), \tag{69}$$

$$\text{Var}_{\theta \sim \mathcal{P}_0}[\Theta_t(x, x') - \Theta_0(x, x')] = \mathcal{O}(n^{-2}). \tag{70}$$

*Proof.* It is enough to show that $\mathbb{E}_{\theta \sim \mathcal{P}_0}[\Theta_1(x, x') - \Theta_0(x, x')] = \mathcal{O}(n^{-1})$. It then follows from Theorem 9 that $\mathbb{E}_{\theta \sim \mathcal{P}_0}[\Theta_{t+1}(x, x') - \Theta_t(x, x')] = \mathcal{O}(n^{-1})$ for all $t$, and therefore

$$\mathbb{E}_{\theta \sim \mathcal{P}_0}[\Theta_t(x, x') - \Theta_0(x, x')] = \sum_{t'=0}^{t-1} \mathbb{E}_{\theta \sim \mathcal{P}_0}[\Theta_{t'+1}(x, x') - \Theta_{t'}(x, x')] = \mathcal{O}(n^{-1}), \tag{71}$$

concluding the proof. We have

$$\mathbb{E}_{\theta \sim \mathcal{P}_0}[\Theta_1(x, x') - \Theta_0(x, x')] = \sum_{\substack{k,l=0 \\ k+l \geq 1}}^{d+1} \frac{(-\eta)^{k+l}}{k!l!} \sum_\mu \mathbb{E}_{\theta \sim \mathcal{P}_0}\left[ \sum_{\mu_1,\dots,\mu_k} \frac{\partial L}{\partial \theta^{\mu_1}} \cdots \frac{\partial L}{\partial \theta^{\mu_k}} T_{\mu\mu_1\dots\mu_k}(x) \right.$$

$$\left. \sum_{\nu_1,\dots,\nu_l} \frac{\partial L}{\partial \theta^{\nu_1}} \cdots \frac{\partial L}{\partial \theta^{\nu_l}} T_{\mu\nu_1\dots\nu_l}(x') \right]. \tag{72}$$

The fact that the $k, l$ sums are truncated at $d + 1$ follows from using linear activations, as higher-order derivatives of the network function vanish in this case. All terms in the sum over $k, l$ include a tensor product of the form $\sum_{\mu,\vec{\mu},\vec{\nu}} T_{\mu\mu_1\dots\mu_k}(x)T_{\mu\nu_1\dots\nu_l}(x')(\cdots)$ with either $k \geq 1$ or $l \geq 1$, where $(\cdots)$ stands for additional derivative tensor factors. Therefore, all terms in the $k, l$ sum are correlation functions that have a cluster of size at least 3, including $T(x)$, $T(x')$, and at least one other tensor contracted through the $\mu_1$ or $\nu_1$ index. It follows from the Conjecture that each term in the sum is $\mathcal{O}(n^{-1})$. Lemma 6 then implies that the variance of these updates is $\mathcal{O}(n^{-2})$. $\qquad \square$

## E.2 WIDE NETWORK EVOLUTION IS LINEAR IN THE LEARNING RATE

In this section we prove that, at large width, the NTK determines the evolution of the network function not just for continuous-time gradient descent but also for discrete-time gradient descent. A similar result holds for stochastic gradient descent, using a stochastic kernel.[11] Again we prove the deep linear case explicitly, but the result holds for deep networks with smooth non-linear activations under the additional assumption that the large width limit and Taylor series can be exchanged.

---

[11]The perspective presented here helps understand the results of (Lee et al. (2019)) where it was observed empirically that linearized evolution is a good description of wide networks even for relatively large learning rates.

**Theorem 12.** *Let $f(x)$ be the network output of a deep linear network, and let $f_t(x)$ be the evolved function after $t$ gradient descent steps, defined by $\theta_{t+1} = \theta_t - \eta \nabla L(\theta_t)$. In the large width limit, each gradient descent step update of $f_t$ is linear in the learning rate $\eta$. Furthermore,*

$$\mathbb{E}_\theta \left[ f_{t+1}(x) - f_t(x) \right] = -\eta \sum_{(x',y') \in D_{\mathrm{tr}}} \mathbb{E}_\theta \left[ \Theta_0(x,x') \frac{\partial \ell_t(x',y')}{\partial f} \right] + \mathcal{O}(n^{-1}). \tag{73}$$

*Here, $\Theta_0$ is the NTK at initialization and $\ell_t$ is the single-sample loss at time $t$.*

*Proof.* For a deep linear network, under a single gradient descent step we have

$$\mathbb{E}_\theta \left[ f_{t+1}(x) - f_t(x) \right] = \sum_{k=1}^{d+1} \frac{(-\eta)^k}{k!} \sum_{\mu_1,\dots,\mu_k} \mathbb{E}_\theta \left[ \frac{\partial^k f_t(x)}{\partial \theta^{\mu_1} \cdots \partial \theta^{\mu_k}} \frac{\partial L_t}{\partial \theta^{\mu_1}} \cdots \frac{\partial L_t}{\partial \theta^{\mu_k}} \right] \tag{74}$$

$$= -\eta \sum_{(x',y') \in D_{\mathrm{tr}}} \mathbb{E}_\theta \left[ \Theta_t(x,x') \frac{\partial \ell_t(x',y')}{\partial f} \right]$$

$$+ \sum_{k=2}^{d+1} \frac{(-\eta)^k}{k!} \sum_{\mu_1,\dots,\mu_k} \mathbb{E}_\theta \left[ \frac{\partial^k f_t(x)}{\partial \theta^{\mu_1} \cdots \partial \theta^{\mu_k}} \frac{\partial L_t}{\partial \theta^{\mu_1}} \cdots \frac{\partial L_t}{\partial \theta^{\mu_k}} \right]. \tag{75}$$

First, consider the sum over $k$ in (73). Each term in the sum is a correlation function for which the cluster graph contains a connected subgraph of size at least 3, and is therefore $\mathcal{O}(n^{-1})$ by Lemma 7 and Theorem 1. In the remaining $\mathcal{O}(\eta)$ term, by the same argument as Theorem 11 we can replace $\Theta_t = \Theta_0 + \mathcal{O}(n^{-1})$ in the correlation function. The result is equation (73). $\qquad\square$

As mentioned above, we note that the proof goes through when using stochastic gradient descent updates, with the difference that in (73) we should sum over mini-batch samples instead of over the entire training set.

### E.3 Spectral properties of the NTK and the Hessian

With an eye towards understanding the structure of the loss landscape at large width and as another example use case of our approach, we investigate the relation between the spectra of the Hessian, and the NTK.

Among other observations, we present an argument that for a network with $d$ hidden layers and MSE loss, the top $D_{\mathrm{in}}$ ordered eigenvalues of the Hessian, $\lambda_i^{(H)}$, are related to those of the kernel $\lambda_i^{(\Theta)}$ by

$$\mathbb{E}_\theta \left[ \lambda_i^{(H)} - \lambda_i^{(\Theta)} \right] = \begin{cases} \mathcal{O}(n^{-1/2}) & , d = 1 \\ \mathcal{O}(n^{-1}) & , d > 1 \end{cases}. \tag{76}$$

The remaining Hessian eigenvalues vanish at large width as $\mathcal{O}(n^{-1/2})$ for one hidden layer networks and $\mathcal{O}(n^{-1})$ for deeper networks.[12] This is experimentally corroborated in Figure 9.

The Hessian of a general loss takes the form,

$$H_{\mu\nu} = \sum_{(x,y) \in D_{\mathrm{tr}}} \Big[ \underbrace{\frac{\partial^2 \ell(x,y)}{\partial f^2} \frac{\partial f(x)}{\partial \theta^\mu} \frac{\partial f(x)}{\partial \theta^\nu}}_{\mathcal{A}_{\mu\nu}} + \underbrace{\frac{\partial \ell(x,y)}{\partial f} \frac{\partial^2 f(x)}{\partial \theta^\mu \partial \theta^\nu}}_{\mathcal{B}_{\mu\nu}} \Big]. \tag{77}$$

The moments of $H$ are a useful way to understand its spectrum. As in other examples that we have seen, traces of powers of $\mathcal{B}$ lead to connected diagrams with higher and higher numbers of vertices, and so scale as increasingly negative powers of $n$. More explicitly, traces involving powers of $\mathcal{B}$ contain multiple contracted derivative tensors. As a result, taking the average of these traces over the

---

[12] Empirically we actually find the even stronger bound $\mathbb{E}_\theta \left[ \lambda_i^{(H)} - \lambda_i^{(\Theta)} \right] = \mathcal{O}(n^{-1})$ for the top $D_{\mathrm{in}}$ eigenvalue differences and $\mathcal{O}(n^{-1/2})$ for the remaining eigenvalues in the one hidden layer case. We can gain insight into this improved scaling through the perspective of degenerate eigenvalue perturbation theory (Trefethen & Bau (1997)), but this is outside the scope of the current presentation.

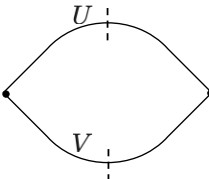

Figure 8: A prototypical diagram corresponding to $\mathbb{E}_\theta \left[ \text{Tr} \left( \mathcal{B}^2 \right) \right]$

weight distribution leads to correlation functions where the associated cluster graph, $G_C$, contain subgraphs with at least three connected vertices. By Lemma 7 and Conjecture 1 these correlation functions vanish at infinite width. For example $\mathbb{E}_\theta \left[ \text{Tr} \left( \mathcal{AB} \right) \right] = \mathcal{O}(1/n)$. There is a single exception to this, as can be easily seen in figure 8, $\mathbb{E}_\theta \left[ \text{Tr}(\mathcal{B}^2) \right]$ is $\mathcal{O}(1)$. Thus most moments of the Hessian are equal to moments of $\mathcal{A}$ in expectation.[13]

$$\mathbb{E}_\theta \left[ \text{Tr} \left( H^m \right) \right] = \left\{ \begin{array}{ll} \mathbb{E}_\theta \left[ \text{Tr} \left( \mathcal{A}^m \right) \right] & , m \neq 2 \\ \mathbb{E}_\theta \left[ \text{Tr} \left( \mathcal{A}^2 \right) \right] + \mathbb{E}_\theta \left[ \text{Tr} \left( \mathcal{B}^2 \right) \right] & , m = 2 \end{array} \right. + \mathcal{O}(1/n). \tag{78}$$

What's more, we can relate moments of $\mathcal{A}$ to those of the kernel, as both are built out of two logit derivatives. Explicitly,

$$\text{Tr}(\mathcal{A}^m) = \text{Tr}((M\Theta)^m), \tag{79}$$

and so the moments of the Hessian are also related to those of the kernel.[14]

$$\mathbb{E}_\theta \left[ \text{Tr} \left( H^m \right) \right] = \left\{ \begin{array}{ll} \mathbb{E}_\theta \left[ \text{Tr} \left( (M\Theta)^m \right) \right] & , m \neq 2 \\ \mathbb{E}_\theta \left[ \text{Tr} \left( (M\Theta)^2 \right) \right] + \mathbb{E}_\theta \left[ \text{Tr} \left( \mathcal{B}^2 \right) \right] & , m = 2 \end{array} \right. + \mathcal{O}(1/n). \tag{80}$$

Here, we have defined,

$$M(x_a, x_b) = \delta_{ab} \frac{\partial^2 \ell(x_a, y_a)}{\partial f^2} \quad : (x_a, y_a) \in D_{\text{tr}}. \tag{81}$$

For the case of MSE loss, we can go even further. In that case, $M(x_a, x_b) = \delta_{ab}$ and $\mathcal{B}$ decays to zero during training. We thus have,

Initially: $$\mathbb{E}_\theta \left[ \text{Tr} \left( H^m \right) \right] = \left\{ \begin{array}{ll} \mathbb{E}_\theta \left[ \text{Tr} \left( \Theta^m \right) \right] & , m \neq 2 \\ \mathbb{E}_\theta \left[ \text{Tr} \left( \Theta^2 \right) \right] + \mathbb{E}_\theta \left[ \text{Tr} \left( \mathcal{B}^2 \right) \right] & , m = 2 \end{array} \right. + \mathcal{O}(1/n) \tag{82}$$

At late times: $\mathbb{E}_\theta \left[ \text{Tr} \left( H^m \right) \right] \to \mathbb{E}_\theta \left[ \text{Tr} \left( \Theta^m \right) \right] \quad \forall m.$

These results indicate that the only difference between the spectra of the Hessian and the NTK come from $\mathcal{B}$ and that $\mathcal{B}$ must have eigenvalues which scale as $1/\sqrt{\dim(\mathcal{B})}$. As $\dim(B) = \mathcal{O}(n)$ for one hidden layer networks and $\mathcal{O}(n^2)$ for deep networks we are left with the relation (76) between the eigenvalues of $\Theta$ and $H$. As the network trains, the difference between these eigenvalues gets even smaller. These results are confirmed experimentally in Figure 9 and Figure 10.

**Multi-class.** This story can be generalized to multi-class neural network maps. In this case the logit is a map $f^A : \mathbb{R}^{D_{\text{in}}} \to \mathbb{R}^{N_{\text{class}}}$, and the NTK has two class indices.

$$\Theta^{AB}(x, x') = \frac{\partial f^A(x)}{\partial \theta^\mu} \frac{\partial f^B(x')}{\partial \theta^\mu}. \tag{83}$$

Expression (80) relating the moments of the Hessian to the NTK still holds in this context provided we take $\text{Tr} \left( (M\Theta)^m \right) \to \text{Tr} \left( (M_{AB}\Theta^{AB})^m \right)$, with the more general matrix,

$$M_{AB}(x_a, x_b) = \delta_{ab} \frac{\partial \ell(x_a, y_a)}{\partial f^A \partial f^B} \quad : (x_a, y_a) \in D_{\text{tr}}. \tag{84}$$

---

[13]For the first moment in linear or ReLU networks, $\text{Tr}\,\mathcal{B} = 0$.

[14]Here we have argued for relations relating the mean of moments. It is not too difficult to see that these relations will also hold for typical realizations. This follows from Lemma 6.

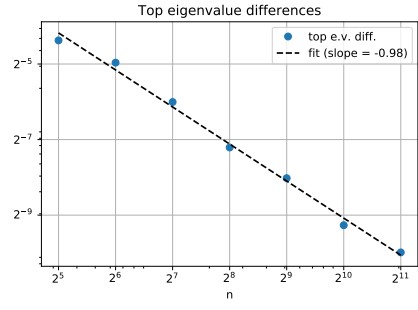

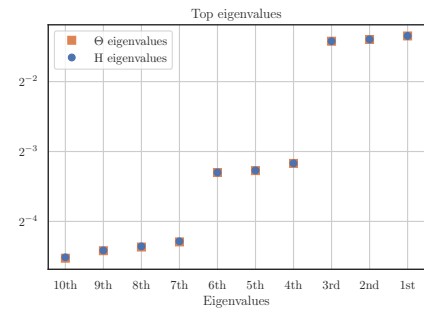

(a) Hessian and NTK e.v. difference.

(b) Degenerate spectrum.

Figure 9: (a) The mean difference between the top eigenvalues of the NTK and the Hessian at initializations for a two hidden layer ReLU network of varying width match well with the predicted $\mathcal{O}(n^{-1})$ behavior. The mean is taken over 100 instances per width on two-class MNIST with 10 images per class. (b) The top 10 eigenvalues of both the Hessian and the NTK for a two-layer ReLU network of width 2048 trained on a **three-class** subset of MNIST. The top eigenvalues match well and show aspects of the repeated $N_{\text{class}}$ structure predicted at large width. The eigenvalues of $H$ are computed using the lanczos algorithm.

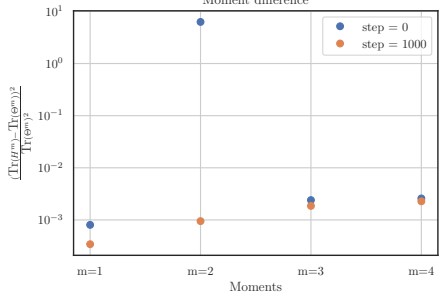

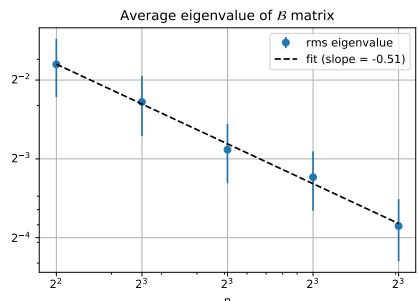

(a) Relative moment differences

(b) Second moment of $\mathcal{B}$

Figure 10: Relation between the spectrum of the Hessian and NTK. (a) The first 4 moments of $H$ and $\Theta$ for two hidden layer ReLU networks. At initialization the first third and fourth moments are numerically similar, while the relative difference of the second moment is $\mathcal{O}(1)$. After training all moments are numerically close. (b) The root mean square eigenvalue of the $\mathcal{B}$ matrix shows good agreement with the predicted $1/\sqrt{n}$ fall off for single hidden layer ReLU networks. Experiments were performed on ReLU networks trained on two-class MNIST with 100 images per class. Moments of the NTK were computed exactly, while moments of the Hessian and $\mathcal{B}$ were computed by randomly sampling 1000 vectors and using Hutchinson's trick.

At large width, the NTK approaches the identity matrix in class space, $\Theta^{AB} \to N_{\text{class}}^{-1} \delta^{AB} \text{Tr}(\Theta)$ (Jacot et al. (2018)). This implies that the NTK spectrum consists of $N_{\text{class}}$ repeated copies. This has consequences for the spectrum of the Hessian at large width. For instance, for the case of MSE error it also implies $N_{\text{class}}$ repeated copies of the Hessian spectrum (See Figure 9b). It is intriguing to think this could serve as a path towards understanding the emergence of the $N_{\text{class}}$ (or $N_{\text{class}} - 1$) large eigenvalues and corresponding subspace observed in the Hessian spectrum (Sagun et al. (2016; 2017); Gur-Ari et al. (2018); Ghorbani et al. (2019); Papyan (2019)).

### E.4 HIGHER-ORDER NETWORK EVOLUTION

In this section we will explain how to compute higher-order corrections to training dynamics. In principle, this prescription allows one to compute model dynamics as an expansion in $1/n$ to arbitrary order. We apply this explicitly to compute the $\mathcal{O}(1/n)$ training dynamics. In Figure 11, we present experimental confirmation of our predictions for the evolution of the NTK. In the main text, we presented these results for the special case of gradient flow with MSE loss.

So far we have mostly been using diagrammatic techniques to understanding the leading order scaling of correlation functions. It is interesting to try and understand finite width networks by asking how training dynamics are modified beyond the leading order asymptotics. There are three clear sources of corrections to the leading behavior.

1. The initial kernel, $\Theta_0$, receives finite width corrections.
2. The linear (in learning rate) update for the network function, equation (73), is modified away from infinite width.
3. The kernel is not constant at finite width.

The first source of corrections is automatically taken into account in typical empirical settings, as the finite-width $\Theta_0$ can be computed explicitly. The other sources of corrections are non-trivial, and will be the focus of this section. We begin by explaining how to take into account the non-constancy of the kernel order by order in $1/n$, while maintaining the continuous time approximation. Next, we back away from the continuous time limit and write down the full discrete evolution. We introduce a method to compute arbitrary order corrections, and explicitly work out the corrections at order $1/n$ for MSE loss.

### E.4.1 CONTINUOUS TIME.

Continuous time evolution of the model function in neural networks is governed by the differential equation

$$\frac{df(x;t)}{dt} = -\sum_{(x',y') \in D_{\text{tr}}} \Theta(x,x';t) \frac{\partial \ell(x',y')}{\partial f} \,. \tag{85}$$

As we have discussed at length, in the large width limit, this equation simplifies and $\Theta(t)$ is asymptotically constant (Jacot et al. (2018)). Our goal is to move beyond this leading behavior at large width and solve (85) order-by-order in a $1/n$ expansion. In Section 3 we described how to compute the $\mathcal{O}(n^{-1})$ corrections for the case of MSE loss. Here we explain how to handle corrections more generally as well as giving a more detailed discussion of the MSE case.

The network map $f$ and kernel, $\Theta$ are members of the following family of operators.

$$O_s(x_1, \ldots, x_s; t) := \sum_{\mu} \frac{\partial O_{s-1}(x_1, \ldots, x_{s-1}; t)}{\partial \theta_\mu} \frac{\partial f(x_s; t)}{\partial \theta_\mu} \,, \quad s \geq 2 \,, \tag{86}$$

with $O_1 := f$. Here, as above, $O_2 = \Theta$ is the kernel. With a general loss function, these operators satisfy.

$$\frac{dO_s(x_1, \ldots, x_s; t)}{dt} = -\sum_{(x',y') \in D_{\text{tr}}} O_{s+1}(x_1, \ldots, x_s, x'; t) \frac{\partial \ell(x',y';t)}{\partial f} \,, \quad s \geq 1 \,. \tag{87}$$

Equations (86) and (87). Give an infinite tower of first order ODEs, the solution of which gives the time evolution of the network map and the kernel.

$$\frac{df(x_1;t)}{dt} = -\sum_{(x,y)\in D_{\text{tr}}} \Theta(x_1,x;t)\frac{\partial \ell(x,y)}{\partial f}$$

$$\frac{d\Theta(x_1,x_2;t)}{dt} = -\sum_{(x,y)\in D_{\text{tr}}} O_3(x_1,x_2,x;t)\frac{\partial \ell(x,y)}{\partial f}$$

$$\frac{dO^{(1)}(x,x',x'';t)}{dt} = -\sum_{(x,y)\in D_{\text{tr}}} O_4(x_1,x_2,x_3,x;t)\frac{\partial \ell(x,y)}{\partial f} \tag{88}$$

$$\vdots$$

Solving this infinite tower is not feasible. If we wish to work to a given order in an expansion in $1/n$, however, there is a dramatic simplification, which makes a solution possible. Firstly, we can truncate these equations. To see this note that the operators $O_s$ contain $s$ contracted derivative tensors. As a result, by Lemma 7 and Conjecture 1, correlation functions involving the operators, $O_s$ satisfy

$$\mathbb{E}_\theta\left[O_s(x_1,\ldots,x_s;t)F(\vec{x};t)\right] = \begin{cases} \mathcal{O}\left(n^{\frac{2-s}{2}}\right) & ,s \text{ even} \\ \mathcal{O}\left(n^{\frac{1-s}{2}}\right) & ,s \text{ odd} \end{cases}, \tag{89}$$

where $F(\vec{x};t)$ is arbitrary additional contribution to the integrand.

Thus, if we wish to work to solve for the time evolution up to corrections which scale as $\mathcal{O}(n^{-r})$ we can truncate the tower at $s = 2r$ and set $O_{2r}(t)$ to be equal to its initial value. Note that the leading order solution, (85), is the result of this procedure with $r = 1$ and the results presented in the main text for the $\mathcal{O}(n^{-1})$ evolution correspond to $r = 2$.

The truncation provides a dramatic simplification, however it is not immediately clear how to solve even the truncated differential equations in (87). We now describe how to organize the perturbative expansion of the operators $O_s(t)$ (including $f$ and $\Theta$) in such a way that the differential equations become tractable.

The central idea is to write each operator, $O_s(t)$, as an expansion.

$$O_s(x_1,\ldots,x_s;t) = \sum_{r=\lfloor\frac{s-1}{2}\rfloor}^{\infty} O_s^{(r)}(x_1,\ldots,x_s;t) \tag{90}$$

where each order $O_s^{(r)}$ captures the $\mathcal{O}(n^{-r})$ evolution of $O_s$. For example,

$$f(x_1;t) = f^{(0)}(x_1;t) + f^{(1)}(x_1;t) + \cdots$$
$$\Theta(x_1,x_2;t) = \Theta^{(0)}(x_1,x_2;t) + \Theta^{(1)}(x_1,x_2;t) + \cdots$$
$$O_3(x_1,x_2,x_3;t) = O_3^{(1)}(x_1,x_2,x_3;t) + O_3^{(2)}(x_1,x_2,x_3;t) + \cdots \tag{91}$$
$$\vdots$$

The notation $O_s^{(r)}$ means both that any correlation function containing $O_s^{(r)}$ is $\mathcal{O}(n^{-r})$ and, by Lemma 6, that typical realizations of the operators scale as $\mathcal{O}(n^{-r})$.[15]

Once the operators $O_s$ are organized in this way, solving the differential equations (85) is tractable. As the differential equations describing the evolution of $O_s^{(r)}$ for $r > 0$ only depend on the time dependent solutions of $O_s^{(r-1)}$, we can iteratively solve for the $O_s^{(r)}$ order by order. For example, in Section 3, we used the leading order solution for $f$ and $\Theta$ to solve for $\Theta^{(1)}$.

In principle this procedure can be extended to arbitrary order in $1/n$. Before going onto explain the finite step corrections to this procedure, we reproduce the results of section 3 in more detail.

---

[15] Note that in Section 3 we used the alternate notation $\Theta_1$ for $\Theta^{(1)}$.

**MSE continuous time example.** The MSE loss is,

$$L^{\text{MSE}} = \frac{1}{2} \sum_{(x,y) \in D_{\text{tr}}} (f(x) - y_a)^2 \tag{92}$$

As such the update equation simplifies to,

$$\frac{df(x;t)}{dt} = - \sum_{(x',y') \in D_{\text{tr}}} \Theta(x, x'; t) \left( f(x'; t) - y' \right) , \tag{93}$$

The leading order solution to equation (93) is exponential, kernel evolution,

$$f^{(0)}(t) = y + e^{-t\Theta_0}(f_0 - y) \tag{94}$$

$$\Theta^{(0)}(t) = \Theta_0 . \tag{95}$$

Here we are using a condensed notation where $f_0 := f^{(0)}(0)$, $\Theta_0 := \Theta^{(0)}(0)$. Equation (94) is a vector equation with $f_0$, $f^{(0)}(t)$, and $y$ are vectors over the training set and the leading order kernel $\Theta_0$ is a square matrix over the same space.

As discussed above to study the $\mathcal{O}(1/n)$ evolution, we can truncate the set of equations in (87) at $s = 4$, and set $O_4(t) = O_4^{(1)}(t) = O_4(0)$ up to corrections which scale as $\mathcal{O}(n^{-2})$.

Moving up a rung in (87), we have

$$\frac{dO_3(x_1, x_2, x_3; t)}{dt} = - \sum_{(x,y) \in D_{\text{tr}}} O_4(x_1, x_2, x_3, x; t) \left( f(x; t) - y \right) . \tag{96}$$

To solve for $O_3^{(1)}(t)$, i.e. neglecting $\mathcal{O}(n^{-2})$ corrections, we can plug in the leading order approximations, to $O_4(t)$ and $f(t)$.

$$\frac{dO_3^{(1)}(x_1, x_2, x_3; t)}{dt} = - \sum_{(x,y) \in D_{\text{tr}}} O_4(x_1, x_2, x_3, x; 0) \left( f^{(0)}(x; t) - y \right) . \tag{97}$$

This gives,

$$O_3^{(1)}(x_1, x_2, x_3; t) = O_3(x_1, x_2, x_3; 0) - \int_0^t dt' \sum_{(x,y) \in D_{\text{tr}}} O_4(x_1, x_2, x_3, x; 0) \left( f^{(0)}(x; t) - y \right) . \tag{98}$$

Using the explicit form of $f^{(0)}(t)$ we can write,

$$O_3^{(1)}(\vec{x}; t) = O_3(\vec{x}; 0) - \sum_a O_4(\vec{x}; 0) \cdot \left( \Theta_0^{-1}(1 - e^{-t\Theta_0})(f_0 - y) \right) . \tag{99}$$

Here we are again using a condensed notation. $\vec{x} = (x_1, x_2, x_3)$. $f_0$ and $y$ are vectors over the training set while $\Theta_0$ is a square matrix, and $O_4(\vec{x}; 0)$ is also a vector over the training set, with the value $O_4(\vec{x}, x'; 0)$ on the point $x' \in D_{\text{tr}}$.

Moving one more step up the ladder gives

$$\Theta(x_1, x_2; t) = \Theta_{x_1, x_2; 0} + \Theta^{(1)}(x_1, x_2; t) + \mathcal{O}(n^{-2}) ,$$

$$\Theta^{(1)}(x_1, x_2; t) = - \int_0^t dt' \sum_{(x,y) \in D_{\text{tr}}} O_3^{(1)}(x_1, x_2, x; t') \left( f^{(0)}(x; t') - y \right) . \tag{100}$$

Plugging in $O_3^{(1)}$, and using the eigen-decomposition of $\Theta_0$ to perform the integrals gives,

$$\Theta^{(1)}(\vec{x}; t) = - \sum_i \frac{1}{\lambda_i} (O_3(\vec{x}; 0) \cdot \hat{e}_i)(\Delta f_0 \cdot \hat{e}_i) \left[ 1 - e^{-t\lambda_i} \right]$$

$$+ \sum_{ij} \frac{1}{\lambda_j} (\hat{e}_i^T O_4(\vec{x}; 0) \hat{e}_j)(\Delta f_0 \cdot \hat{e}_i)(\Delta f_0 \cdot \hat{e}_j) \left[ \frac{1 - e^{-t\lambda_i}}{\lambda_i} - \frac{1 - e^{-t(\lambda_i + \lambda_j)}}{\lambda_i + \lambda_j} \right] . \tag{101}$$

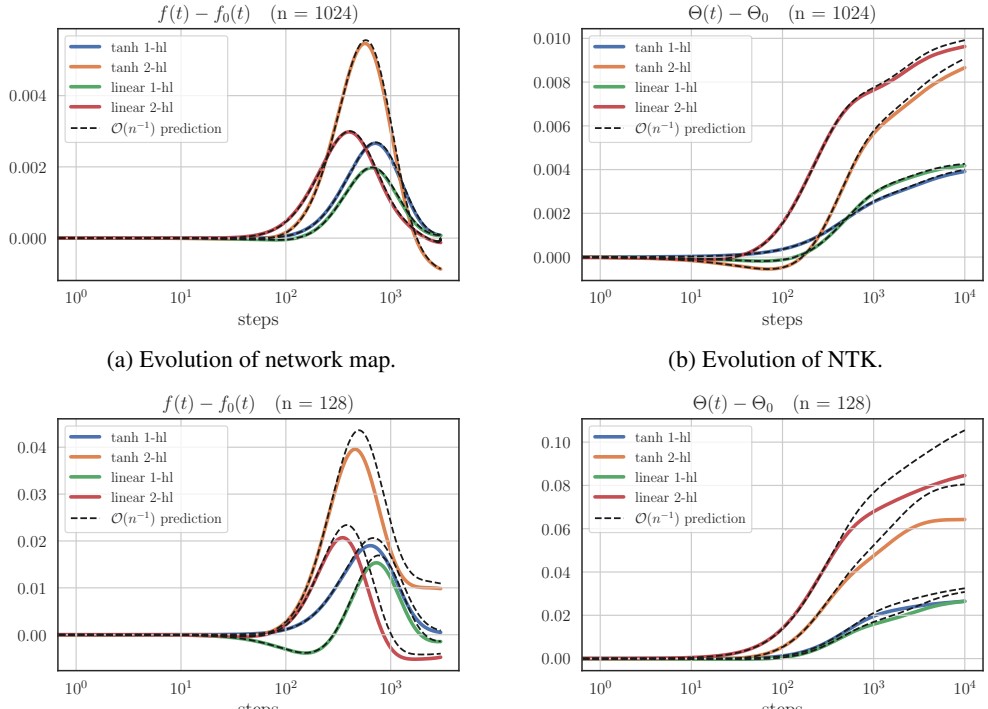

(a) Evolution of network map.

(b) Evolution of NTK.

(c) Evolution of network map for a narrower network.

(d) Evolution of NTK for a narrower network.

Figure 11: Single instance evolution for a randomly selected element of the NTK and network map. Theoretical predictions at $\mathcal{O}(n^{-1})$ are represented by dashed lines, while the true network evolution is shown in solid. (a-b) Wide networks show excellent agreement with predictions. (c-d) Predictions match reasonably well even down to modest widths. All plots correspond to single training runs on 2-class MNIST with 10 samples per class. The learning rate is 0.1 for all runs. All runs reach training accuracy 1.

Here, we have introduced the eigenvectors, $\hat{e}_i$, which are vectors over the training dataset and eigenvalues $\lambda_i$ of the initial kernel, $\Theta_0$. The vector $\vec{x} = (x_1, x_2)$ as $\Theta$ depends on two inputs. $O_3(\vec{x}; 0)$ is a vector over the dataset while $O_4(\vec{x}; 0)$ is a square matrix and $\Delta f_0 := f_0 - y$ is a vector over the training data.

Finally, we can plug in the expression (100) into (93) and give the sub-leading behavior for $f(t)$

$$\frac{df(x; t)}{dt} = - \sum_{(x, y) \in D_{\text{tr}}} \left( \Theta(x, x'; 0) + \Theta^{(1)}(x, x'; t) \right) (f(x'; t) - y) + \mathcal{O}(n^{-2}) \tag{102}$$

$$f(t) = y + e^{-\Theta_0 t} \left( 1 - \int_0^t dt' e^{t' \Theta_0} \Theta^{(1)}(t') e^{-t' \Theta_0} \right) (f_0 - y) . \tag{103}$$

This completes the full $\mathcal{O}(n^{-1})$ time dependance. This prediction for the $\mathcal{O}(1/n)$ time dependence is confirmed experimentally in Figure 11.

Note, one consequence of (101) is that the $\mathcal{O}(1/n)$ corrections to $\Theta$ go to a constant at late times.

$$\Theta_\infty^{(1)} := \lim_{t \to \infty} \Theta^{(1)}(t) = - \sum_i \frac{1}{\lambda_i} (O_3(\vec{x}; 0) \cdot \hat{e}_i)(\Delta f_0 \cdot \hat{e}_i) + \sum_{ij} \frac{1}{\lambda_i(\lambda_i + \lambda_j)} (\hat{e}_i^T O_4(\vec{x}; 0) \hat{e}_j)(\Delta f_0 \cdot \hat{e}_i)(\Delta f_0 \cdot \hat{e}_j) . \tag{104}$$

And the asymptotic evolution of $\Delta f(t)$ is again constant kernel evolution, but with a corrected kernel.

$$f(t) \to y + e^{-t(\Theta_0 + \Theta_\infty^{(1)})} (f_0 - y) . \tag{105}$$

It is worth noting that these predictions are quite detailed, giving the full time dependence, rather than just the overall scaling with width. In deriving these results and in the discrete time analysis below, we implicitly make an assumption that we can control the discrete time corrections or in the continuous time case that $f(t)$, $\Theta(t)$ and all the $O_s(t)$ are differentiable. This relies on being able to differentiate through the activation function. For non-smooth activations such as ReLU, this assumption is suspect, and indeed the evolution is more subtle in the ReLU case and a full analysis is left to future work. For this reason we present experimental results for networks with smooth activation functions.

### E.4.2   DISCRETE TIME.

With the $1/n$ corrections to continuous time evolution under our belts, it is possible to also keep track of corrections coming from the discrete update step. The essential point, that it is possible to solve iteratively, order by order in $1/n$ is not changed. To see this we can look at how the update equations are modified. Beginning with the network output update equation, we have,

$$
f_{t+1}(x) - f_t(x) = -\eta \sum_{(x',y') \in D_{\mathrm{tr}}} \Theta_t(x,x') \frac{\partial \ell_t(x',y')}{\partial f}
$$
$$
+ \frac{\eta^2}{2} \sum_{\mu\nu} \frac{\partial^2 f_t(x)}{\partial \theta^\mu \partial \theta^\nu} g_t^\mu g_t^\nu - \frac{\eta^3}{3!} \sum_{\mu\nu\rho} \frac{\partial^3 f_t(x)}{\partial \theta^\mu \partial \theta^\nu \partial \theta^\rho} g_t^\mu g_t^\nu g_t^\rho + \cdots . \tag{106}
$$

In Section E.2 we argued that the $\mathcal{O}(\eta^2)$ terms vanish at large width. In more detail, each factor of the gradient, $g^\mu$, contains a contracted derivative tensor. Thus, the cluster graph for correlation functions involving the $\eta^2$ term will always contain three contracted vertices and thus the correlation functions will scale as $\mathcal{O}(n^{-1})$. The $\eta^3$ term will give four vertices, and thus also scale as $\mathcal{O}(1/n)$, the $\eta^4$ contribution will be $\mathcal{O}(1/n^2)$ and so on.

Just as in the continuous time case this equation is still solvable order by order in $1/n$. What we mean by solvable is the following. All terms on the RHS appearing with $f_t$ are already higher order as a result of their derivative structure. This means, just as in the continuous case, we can use the lower order solution of $f_t$ in the gradient terms on the RHS to solve for $f_t$ on the LHS.

Similarly, the update equation for $\Theta$ receives discrete time modifications.

$$
\Theta_{t+1}(x_1,x_2) - \Theta_t(x_1,x_2) = -\eta \sum_a O_{3;t}(x_1,x_2,x) \frac{\partial \ell_t(x,y)}{\partial f} + \frac{\eta^2}{2} \sum_{\mu\nu} \frac{\partial^2 \Theta_t(x_1,x_2)}{\partial \theta^\mu \partial \theta^\nu} g_t^\mu g_t^\nu + \cdots .
$$
$$
\tag{107}
$$

Here we have adopted a notation where, $\vec{x} = (x_1, \ldots, x_s)$. Just as for $f$, increased powers in $\eta$ are increasingly suppressed in $n$ and we can iteratively solve this equation using solutions at leading orders in $n$ to solve for sub-leading behavior.

More generally the tower of differential equations defined recursively in (87) becomes

$$
O_{s;t+1}(\vec{x}) - O_{s;t}(\vec{x}) = - \sum_{(x',y') \in D_{\mathrm{tr}}} O_{s+1;t}(\vec{x},x') \frac{\partial \ell(x',y';t)}{\partial f}
$$
$$
+ \sum_{k=2}^{\infty} \frac{\eta^k}{k!} \sum_{\mu_1,\mu_2,\ldots,\mu_k} \frac{\partial^k O_{s;t+1}(\vec{x})}{\partial \theta_{\mu_1} \partial \theta_{\mu_2} \ldots \partial \theta_{\mu_k}} g_{\mu_1} g_{\mu_2} \cdots g_{\mu_k} , \quad s \geq 1 . \tag{108}
$$

As in the continuous time case, at a given order in $n$, we can truncate this tower and use lower order solutions to solve for the time evolution up to the desired order. To ground this discussion in a concrete use case, we again walk through the $1/n$ corrections for MSE loss, now in discrete time.

**MSE discrete time example.**

For discrete time evolution, the leading order behavior of the model map is

$$
f_t^{(0)} = (1 - \eta\Theta_0)^t (f_0 - y) \tag{109}
$$
$$
\Theta_t^{(0)} = \Theta_0 . \tag{110}
$$

At next order we can proceed by solving the truncated set of equations in (108). The first equation, for $O_{4;t}$ still gives a constant solution,

$$O_{4;t}^{(1)} = O_{4;0} \,. \tag{111}$$

Here, both the discrete and continuous time terms that would appear on the RHS are $\mathcal{O}(1/n^2)$.

Next, we must solve for $O_{3;t}$. The discrete time update is,

$$O_{3;t+1}(\vec{x}) - O_{3;t}(\vec{x}) = -\eta \sum_{(x',y') \in D_{\mathrm{tr}}} O_{4;t}(\vec{x}, x') \left(f_t(x') - y'\right) + \frac{\eta^2}{2} \sum_{\mu\nu} \frac{\partial^2 O_{3;t}(\vec{x})}{\partial\theta^\mu \partial\theta^\nu} g_t^\mu g_t^\nu + \cdots \,. \tag{112}$$

To order $1/n$ we can drop the order $\eta^2$ and higher terms, as they contain expressions with 5 or more contracted derivative tensors. Thus at this order, we are left with the discrete analogue of (96).

$$O_{3;t+1}(\vec{x}) - O_{3;t}(\vec{x}) = - \sum_{(x',y') \in D_{\mathrm{tr}}} O_{4;t}(\vec{x}, x') \left(f_t(x') - y'\right) \,. \tag{113}$$

which we can sum to get the discrete version of (98),

$$O_{3;t}^{(1)}(\vec{x}) = O_{3;0}(\vec{x}) - \eta \sum_{t'=1}^{t} \sum_{(x',y') \in D_{\mathrm{tr}}} O_{4;0}(\vec{x}, x') \left(f_{0;t}(x') - y'\right) \,. \tag{114}$$

Explicitly, this takes the form

$$O_{3;t}^{(1)}(\vec{x}) = O_{3;0}(\vec{x}) - O_{4;0}(\vec{x}) \cdot \left(\Theta_0^{-1}(1 - \eta\Theta_0)(1 - (1 - \eta\Theta_0)^t)(f_0 - y)\right) \,. \tag{115}$$

Here we have adopted notation similar to above, where $O_{4;0}(\vec{x})$ is a vector over the training dataset.

So far, this procedure has been a discrete analogue of what we have done in the continuous time case, however as we move on to compute $\Theta$ and $f$ we will have to keep track of the novel corrections, which vanish in continuous time. Explicitly, at order $1/n$ the discrete update for $\Theta_t$ is given by,

$$\Theta_{t+1}^{(1)}(\vec{x}) - \Theta_t^{(1)}(\vec{x}) = -\eta \sum_a O_{3;t}^{(1)}(\vec{x}, x') \left(f_t^{(0)}(x') - y'\right)$$
$$+ \frac{\eta^2}{2} \sum_{x', x'', y', y'' \in D_{\mathrm{tr}}} \sum_{\mu\nu} \frac{\partial^2 \Theta_0(\vec{x})}{\partial\theta^\mu \partial\theta^\nu} \frac{\partial f_0(x')}{\partial\theta^\mu} \frac{\partial f_0(x'')}{\partial\theta^\nu} \left(f_t^{(0)}(x') - y'\right) \left(f_t^{(0)}(x'') - y''\right) \,. \tag{116}$$

This can be summed to give $\Theta_t^{(1)}$.

To move up to the neural network map itself there is one additional complication. The $\mathcal{O}(\eta^2/n)$ term, $\sum_{\mu\nu} \frac{\partial^2 f_t(x)}{\partial\theta^\mu \partial\theta^\nu} \frac{\partial f_t(x')}{\partial\theta^\mu} \frac{\partial f_t(x'')}{\partial\theta^\nu}$, in (106) has non trivial time dependence. We can deal with this just as we have been doing, by taking an extra time derivative and integrating. Defining,

$$O_{1,1;t}(x_1, x_2, x_2) := \sum_{\mu\nu} \frac{\partial^2 f_t(x_1)}{\partial\theta^\mu \partial\theta^\nu} \frac{\partial f_t(x_2)}{\partial\theta^\mu} \frac{\partial f_t(x_3)}{\partial\theta^\nu} \,. \tag{117}$$

We have

$$O_{1,1;t+1}^{(1)}(\vec{x}) - O_{1,1;t}^{(1)}(\vec{x}) = -\eta \sum_{(x',y') \in D_{\mathrm{tr}}} \sum_\mu \frac{\partial O_{1,1;0}(\vec{x})}{\partial\theta^\mu} \frac{\partial f_t^{(0)}(x')}{\partial\theta^\mu} \left(f_t^{(0)}(x') - y'\right) \,, \tag{118}$$

with the solution,

$$O_{1,1;t}^{(1)}(\vec{x}) = O_{1,1;0}(\vec{x}) - \sum_\mu \frac{\partial O_{1,1;0}(\vec{x})}{\partial\theta^\mu} \frac{\partial f_0}{\partial\theta^\mu} \cdot \left(\Theta_0^{-1}(1 - \eta\Theta_0)(1 - (1 - \eta\Theta_0)^t)\Delta f_0\right)_a \,. \tag{119}$$

This can in turn be plugged into the update equation for $f_t$.

$$
\begin{aligned}
f_{t+1}(x) - f_t(x) = -\eta \sum_a \left( \Theta_0(x, x_a) + \Theta_t^{(1)}(x, x_a) \right) (f_t(x_a) - y_a) \\
+ \frac{\eta^2}{2} \sum_{a,b} O_{1,1;t}^{(1)}(x, x_a, x_b) \left( f_t^{(0)}(x_a) - y_a \right) \left( f_t^{(0)}(x_b) - y_b \right) \\
- \frac{\eta^3}{3!} \sum_{a,b,c} \frac{\partial^3 f_0(x)}{\partial\theta^\mu \partial\theta^\nu \partial\theta^\rho} \frac{\partial f_0(x_a)}{\partial\theta^\mu} \frac{\partial f_0(x_b)}{\partial\theta^\nu} \frac{\partial f_0(x_c)}{\partial\theta^\rho} (f_t(x_a) - y_a)(f_t(x_b) - y_b)(f_t(x_c) - y_c) \, .
\end{aligned}
$$

$$(120)$$

Here $(x_a, y_a)$ are elements of the training set $D_{\mathrm{tr}}$ and summed over. This equation can be solved to give $f_t^{(1)}$.

$$
f_t^{(1)} = (1 - \eta\Theta_0)^t \sum_{t'=1}^t (1 - \eta\Theta_0)^{-(t'+1)} \left[ -\eta\Theta_{\mathrm{NLO}, t'} (1 - \eta\Theta_0)^{t'} \Delta f_0 + \mathrm{Disc}_{t'} \right] \, .
\qquad (121)
$$

Where $\mathrm{Disc}_t$ contains the discrete derivative updates at $\mathcal{O}(1/n)$.

$$
\begin{aligned}
\mathrm{Disc}_t := \frac{\eta^2}{2} \sum_{a,b} O_{1,1;t}^{(1)}(x_a, x_b)(f_t(x_a) - y_a)(f_t(x_b) - y_b) \\
- \frac{\eta^3}{3!} \sum_{a,b,c} \frac{\partial^3 f_0}{\partial\theta^\mu \partial\theta^\nu \partial\theta^\rho} \frac{\partial f_0(x_a)}{\partial\theta^\mu} \frac{\partial f_0(x_b)}{\partial\theta^\nu} \frac{\partial f_0(x_c)}{\partial\theta^\rho}(f_t(x_a) - y_a)(f_t(x_b) - y_b)(f_t(x_c) - y_c) \, .
\end{aligned}
$$

$$(122)$$

These expressions may look fairly intimidating. The key point is that all terms in the summand in (116) and thus (121) are known functions of time and initial data, just as in the continuous time setting.

