# OpenReview forum: "Asymptotics of Wide Networks from Feynman Diagrams"
_ICLR.cc/2020/Conference — Accept (Spotlight)_

### Official Review · AnonReviewer3 · 2019-10-28
**Official Blind Review #3**

**Rating:** 6

**Review:**

The paper investigates the asymptotic behavior of correlation functions of wide networks. The main part of the work revolves around Conjecture 1 which assesses how the correlation function scales depending on the number of connected components of even/odd size of the constructed cluster graph. The authors then study training dynamics of wide networks under gradient flow and (stochastic) gradient descent and present empirical evidence to support their claims.

While I am no expert in this particular area, the paper is fairly well written and the main concepts and ideas are outlined nicely in most places. I do have some comments, though, wrt. the notation and the relevance of certain results:

(1) In 2.1. the authors state the def. of a deep LINEAR network, however, the activation functions are non-linear. Could you be more clear what you mean by linear in your setting?

(2) What are the vertices in Conjecture 1? v_i = T(x_i), right? I think it's clear from the context (i.e., the definition of edges), but this could be written more precisely.

(3) What is the actual relevance of the ORDER of the correlation function in Conjecture 1 and the relevance of the cluster graph. Can the authors motivate this more clearly, or provide some more intuition on this construction?

(4) What does the notation x_1 \leftrightarrow x_2 in Eq. 8 mean?

(5) Thm. 3 - Apparently, the expression for C only depends on the #loops in \gamma. Is this independent of the number of connected components (especially, since Conjecture 1 explicitly hinges on #connected components)?

(6) Does the analysis equally hold if you consider affine maps (Ax + b ) instead of linear operations Ax?

Overall, the paper presents some quite interesting results, but the authors could be more clear on the relevance of those results and its implications. Non-expert readers are left with having to figure this out on their own. In my point of view, this would make the paper much stronger.

**Experience Assessment:**

I do not know much about this area.

**Review Assessment: Checking Correctness Of Derivations And Theory:**

I assessed the sensibility of the derivations and theory.

**Review Assessment: Checking Correctness Of Experiments:**

I assessed the sensibility of the experiments.

**Review Assessment: Thoroughness In Paper Reading:**

I read the paper at least twice and used my best judgement in assessing the paper.

---

> ### Author Response · Authors · 2019-11-12
> **Authors' response**
>
> The authors would like to thank the reviewer for taking the time to review our paper, and for their detailed feedback! We uploaded a revision in which we believe their concerns are addressed. Specifically:
>
> > (1) In 2.1. the authors state the def. of a deep LINEAR network, however, the activation functions are non-linear. Could you be more clear what you mean by linear in your setting?
>
> This was a typo, the word `linear’ should not appear. Fixed in the revised version.
>
> > (2) What are the vertices in Conjecture 1? v_i = T(x_i), right? I think it's clear from the context (i.e., the definition of edges), but this could be written more precisely.
>
> Correct. This is now clarified in the Conjecture.
>
> > (3) What is the actual relevance of the ORDER of the correlation function in Conjecture 1 and the relevance of the cluster graph. Can the authors motivate this more clearly, or provide some more intuition on this construction?
>
> The method presented in this work is a general-purpose tool that can be added to a researcher's toolbox. It allows a researcher to quickly identify ways in which network behavior and training dynamics simplify at large width, for example by deriving which quantities vanish in this limit. As we demonstrate, using this method one is able to dramatically cut down on the amount of effort required to derive several key results related to wide networks, as well as to go beyond the infinite width limit and investigate ordinary neural networks.
> As to the cluster graph, it lets us easily read off the order of a correlation function from the structure of summed derivatives acting on the network map.
> We added the motivation behind studying the order of correlation functions to the introduction.
> > (4) What does the notation x_1 \leftrightarrow x_2 in Eq. 8 mean?
>
> It means “add the same term, except we exchange x_1 and x_2 where they appear”. Now clarified below eq. 8.
>
> > (5) Thm. 3 - Apparently, the expression for C only depends on the #loops in \gamma. Is this independent of the number of connected components (especially, since Conjecture 1 explicitly hinges on #connected components)?
>
> For networks with a single hidden layer, the number of loops in a diagram is equal to the number of connected components. For deep linear networks this is no longer the case, and the quantity that gives a meaningful bound is the number of components, as in Conjecture 1. This is now clarified in Theorem 3.
>
> > (6) Does the analysis equally hold if you consider affine maps (Ax + b ) instead of linear operations Ax?
>
> Yes, the analysis also holds with bias. We added a proof in the appendix (section B.3).

---

### Official Review · AnonReviewer2 · 2019-10-30
**Official Blind Review #2**

**Rating:** 6

**Review:**

This paper proposes a new tool based on Feynman diagrams to analyze wide networks (e.g., feed-forward networks with one or more large hidden layers or CNNs with a large number of filters).

The main contributions of the paper are:
- a new method (using Feynman diagrams) to bound the asymptotic behavior of correlation functions (ensemble averages of the network functions and its derivatives). The method is presented as a conjecture.
- tighter bounds for gradient flow of wide networks
- an extended analysis of SGD for wide networks
- a formalism for deriving finite-width corrections


The study of (infinitely) wide networks has been active over the last few years. A better understanding of wide networks could, amongst other things, shed light on recent empirical results related to over-parametrized networks.  As such, improving our theoretical understanding of wide networks and especially properties of finite-width networks, which is what this paper explores, seems significant and potentially very impactful.

**Experience Assessment:**

I do not know much about this area.

**Review Assessment: Checking Correctness Of Derivations And Theory:**

I assessed the sensibility of the derivations and theory.

**Review Assessment: Checking Correctness Of Experiments:**

I did not assess the experiments.

**Review Assessment: Thoroughness In Paper Reading:**

I read the paper at least twice and used my best judgement in assessing the paper.

---

### Official Review · AnonReviewer4 · 2019-11-03
**Official Blind Review #4**

**Rating:** 8

**Review:**

This is a positive review. Feel free to skip to the feedback.

SUMMARY OF PAPER
This paper explains how to use cluster graphs to easily compute the asymptotic behaviour of any given correlation function (Definition 1) for deep linear networks. By "asymptotic behaviour" I mean that it upper-bounds correlation functions by c·n^s, where s is a nonnegative integer given by the particular cluster graph, and c a "constant" (which I think depends on the particular input, x, to correlation function, among other things).

The authors then conjecture (Conjecture 1) that these bounds transfer to deep nonlinear networks, and that they are tight:
- Appendix C proves that these upper bounds also hold for deep ReLU networks, and 1-hidden-layer networks with smooth nonlinearity. This is also mentioned in page 3.
- Section 2.3 empirically shows that these bounds are pretty tight. (in terms of the exponent, none of the theory here gives a value for the constant c)

The tool provided above is the main result. The authors then use it to provide some results about wide networks
- They give a different proof that for large width, the Neural Tangent Kernel stays constant during training. This is because its derivative wrt. time as a function of width n is O(n^-1), and thus 0 for n->infinity.
- Using the ease of calculation from the tool, they approximate the change in the NTK over training time for any network. They do this using its value at initialization + a term that depends on n^-1. These results are numerically verified in Figure 1.
- They present numerical evidence for the accuracy of this approximation to the change in NTK over time.

The authors spend the last 2 pages explaining how cluster graphs derive from Feynman diagrams (FDs), and why these help compute asymptotics.

WHY I AM ACCEPTING THIS PAPER

The paper adapts FDs and cluster graphs, which is a potentially very useful tool for other wide-network researchers, and could accelerate research in this whole sub-field. It also shows their power by providing a surprisingly large amount of novel theoretical results.

FEEDBACK

At a very high level, there is only one thing that I think isn't made quite clear by the presentation, and it should be. If I understand correctly, Feynman diagrams (or cluster graphs) are only used here to calculate correlation functions for deep *linear* networks. Then, other results establish that the width-dependent asymptotic behaviour for linear networks holds as-is for nonlinear networks, and these results with FDs constitute Conjecture 1. There are proofs for ReLU and 1-layer smooth networks, mentioned in pg. 3; and the experiments in the paper support it for common nonlinear deep networks as well. I think that asymptotics for linear networks transfer to nonlinear ones is an interesting result, which doesn't depend on FDs.

What follows are details.

It is unclear to me whether cluster graphs are as "powerful" as FDs, i.e. whether the bound at the end of the Proof in page 8 is always saturated. Are there some cases in which you need to use FDs to get a tighter upper bound?

In Table 1 you should say that the values under "lin. ReLU tanh" are the fitted exponent s_C. This is not explained. Perhaps you  can mark the only 2 cases (in the 5th row) where the bound is not tight. It would be nice to know how much error remains between the fitted c·n^(s_C) and the empirical values.

Please explain x1 <-> x2 in eq. 8

In figure 1b, consider adding the finite-width limit prediction for the training dynamics. You have already done so for the prediction of the NTK during training in figure 5c, you could indicate it in the same way in 5b.


Typos:

Figure 2 caption: feynman -> Feynman

pg 8. anlytic evicence -> analytic

**Experience Assessment:**

I have published one or two papers in this area.

**Review Assessment: Checking Correctness Of Derivations And Theory:**

I assessed the sensibility of the derivations and theory.

**Review Assessment: Checking Correctness Of Experiments:**

I assessed the sensibility of the experiments.

**Review Assessment: Thoroughness In Paper Reading:**

I read the paper thoroughly.

---

> ### Author Response · Authors · 2019-11-12
> **Authors' response**
>
> The authors would like to thank the reviewer for their detailed and thoughtful feedback! We uploaded a revision that we believe addresses their concerns. All minor corrections and suggested improvements have been taken into account. In addition:
>
> > It is unclear to me whether cluster graphs are as "powerful" as FDs, i.e. whether the bound at the end of the Proof in page 8 is always saturated. Are there some cases in which you need to use FDs to get a tighter upper bound?
>
> The answer to this depends on the activation. For linear and ReLU networks, there are cases in which Feynman diagrams provide a tighter bound than the formula of Conjecture 1. For example, see Table 1, 5th example. The formula from Conjecture 1 predicts an exponent of -1, while a full diagrammatic calculation gives an exponent of -2, which matches the empirical results. However, for tanh activation the formula of Conjecture 1 gives a tight bound in all cases we tested. We explain this in section 2.3 of the revised version.

---

> > ### Comment · AnonReviewer4 · 2019-11-12
> > **I'm confused: I thought FDs were agnostic to activation**
> >
> > > The answer to this depends on the activation. For linear and ReLU networks, there are cases in which Feynman diagrams provide a tighter bound than the formula of Conjecture 1. For example, see Table 1, 5th example. The formula from Conjecture 1 predicts an exponent of -1, while a full diagrammatic calculation gives an exponent of -2, which matches the empirical results.
> >
> > This would answer my question: that FDs do provide tighter bounds than cluster graphs (CGs). However, as I understand it, both CGs and FDs apply only to linear networks. (then, completely separately, you show that the bounds for linear networks apply to some nonlinear ones, and conjecture it for the others).
> >
> > Since FDs and cluster graphs ignore the activation functions, the FD/CG for the tanh, ReLU and identity nonlinearities is the same. Thus it should predict the same upper bound for ReLU, identity and tanh networks with the same architecture. However, if this upper bound is -2, then it is violated by tanh in the experiments in table 1.
> >
> > There also does not seem to be a reason why cluster graphs, instead of FDs, are named in Conjecture 1. As I said repeatedly, from what I understand you get the results that support Conjecture 1 in two independent steps.
> >
> > So at least one of these things is true: Conjecture 1 with FDs substituted in does not apply to tanh networks, or I did not understand the way in which you get the results for Theorems 1 and 2. Could you please help me understand what is going on?

---

> > > ### Author Response · Authors · 2019-11-12
> > > **Clarifying the relationship between Feynman diagrams and cluster graphs**
> > >
> > > We will try to clarify the relationship between activations, Feynman diagrams (FDs), and cluster graphs (CGs). We are happy to add these clarifications to the paper if they are useful!
> > >
> > > FDs and CGs are complementary tools. CGs provide a correct upper bound in all cases we prove, as well as in all cases we tested empirically. FDs can disagree with CGs: FDs give tighter bounds on deep linear networks, but cannot be used to derive bounds for general activations (e.g. tanh). This is one reason the conjecture is phrased in terms of CGs and not FDs.
> > >
> > > Analytically, we show that Feynman diagrams provide upper bounds on the asymptotics of deep linear networks. We further show that cluster graphs provide upper bounds on some networks with non-linearities (as well as on deep linear networks). Cluster graphs are a useful concept even for deep linear networks, because they are much simpler to use than Feynman diagrams. On the other hand, Feynman diagrams are useful because they provide tight bounds for deep linear networks. They are also useful because they are used heavily in the proof of Conjecture 1 for deep linear networks.
> > >
> > > > So at least one of these things is true: Conjecture 1 with FDs substituted in does not apply to tanh networks….
> > >
> > > This is correct. Conjecture 1 with FDs substituted does not apply to deep tanh networks. However, Conjecture 1 as written does apply to tanh networks.

---

### Decision · Program_Chairs · 2019-12-19

**Decision:**

Accept (Spotlight)

**Comment:**

This submission presents bounds on the training dynamics (including gradient evolution) for deep linear (and in some cases nonlinear) networks as a function of the width of the layers or number of convolutional layers. The work also presents experimental results that provide evidence that the bounds are tight.

Strengths:
The work provides interesting insights into these training dynamics, particularly for the wide-but-not-infinite setting, which is less studied.
The work also adapts cluster graphs and Feynman diagrams to derive these bounds, which could be useful tools for researchers in this field.

Weaknesses:
The validity and applicability of some of the results for nonlinear networks was not entirely clear at first but has been clarified in the revision.

The reviewer consensus was to accept this submission.